



# 1 SAO, AO, QBO, and Long-term trend of the peak OH airglow
# 2 emission

Sheng-Yang Gu[1*], Dong Wang[1], Liang Tang[2], Yafei Wei[1]
[1]Electronic Information School, Wuhan University, Wuhan, China.
[2]School of Optoelectronic Engineering, Chongqing University of Posts and Telecommunications, Chongqing, China.
*Corresponding author: Sheng-Yang Gu, (gushengyang@whu.edu.cn)
**Abstract.** Based on the volume emission rate of the OH airglow observed by TIMED/SABER, we fitted the peak emission
rate and the peak height of the OH airglow and analyzed the seasonal and interannual variations of both. The results show
similar latitudinal variations in the semiannual oscillation (SAO) and annual oscillation (AO) of peak emission rate and peak
height: the amplitude of SAO is greatest in equatorial regions and AO is greatest in mid-latitudes. For interannual variations,
we find that OH airglow emission in equatorial regions is modulated by the quasi-biennial oscillation (QBO), while the QBO
signal at other latitudes is much weaker than in equatorial regions and can be ignored. The QBO in OH airglow is consistent
with the phase variation of the QBO in the tropical lower stratosphere (30km), which is also consistent with the phase variation
of the QBO in the migrating diurnal tide. As an important kinetic process affecting OH airglow emission, we suggest that the
tides play an important role in the modulation of the OH airglow by the QBO. In addition, we have analyzed the relationship
between peak OH airglow emission and solar activity. The results show a good correlation between peak emission rate and
solar activity, with a correlation coefficient of 0.89, while peak height shows no significant solar cycle variation, with a
correlation coefficient of -0.66. The modulation of peak emission rate by solar activity has significant latitudinal variation.
The modulation effect is weakest in the equatorial region and greatest at mid-latitudes in both hemispheres.

## 22 1 Introduction

Airglow is the product of photochemical processes in the middle and upper atmosphere, and its radiation intensity is related to
atmospheric temperature and atmospheric density. At the same time, its time and space distribution is modulated by various
atmospheric dynamics processes such as atmospheric gravity waves, tidal waves, and planetary waves. As the radiation of the
atmosphere itself, airglow carries rich information about the middle and upper atmosphere and can be used as a medium to
study various dynamics and photochemical processes in the middle and upper atmosphere. The mesosphere/low thermosphere
(MLT) region, where complex optical phenomena and dynamical processes exist, is a key middle and upper atmosphere study
region. The OH Meinel airglow, one of the most intense airglow emissions in the nightglow, is located near an altitude of



about 87 km. By studying the variation of OH airglow emission, the study of various photochemical and kinetic processes as
well as matter and energy transport processes in the MLT region can be realized.
The changing characteristics of the OH airglow have been studied for a long time. Whether using ground-based observations
or satellite remote sensing observations, there have been many studies on the seasonal variation of OH airglow intensity. Abreu
and Yee (1989) analyzed the nighttime patterns of OH(8-3) airglow measured on the Atmosphere Explorer E satellite and find
that the diurnal variation of hydroxyl emissions is a function of latitude and season. They also observed strong semiannual
oscillation with a maximum near the equator. Takahashi et al. (1995) measured the OH airglow using a ground-based
multichannel airglow photometer at Fortaleza (3.9°S, 38.4°W) and found that the OH airglow intensity shows semiannual
oscillation, with a maximum at the equinoxes and a minimum at the solstices. Zaragoza et al. (2001) used OH airglow data
measured by ISAMS on the URAS satellite and found significant semiannual variations at low latitudes, especially at equatorial
regions, and annual variations at higher latitudes. Lopez-Gonzalez et al. (2004) analyzed more than 3 years of OH(6-2) airglow
observations at the Sierra Nevada Observatory (37.06°N, 3.38°W) and found that the amplitudes of annual and semiannual
variations in OH emission rates were comparable. Buriti et al. (2004) analyzed the intensity data of OH(6,2) airglow radiation
observed at equatorial stations from 1998 to 2001 and found a half-yearly cycle of variation. The maximum value of airglow
intensity appears at the equinoxes and the minimum value appears at the solstices. They suggest that atmospheric tidal
oscillations play an important role in the observed semiannual oscillation. Taylor et al. (2005) found semiannual oscillations
in mesospheric temperature and the intensity of the OH(6,2) and O2(0,1) airglow using 25 months of Maui (20.8°N, 156.2°W)
mesopause thermometer (MTM) observations, and that the spring perturbation was also greater than in autumn. Shepherd et
al. (2006) analyzed the seasonal variations of O(1S) and OH(8,3) nightglow from data obtained by the Wind Imaging
Interferometer on UARS (WINDII) and found a strong semiannual variation dominating at the equator, which is thought to be
driven by the semiannual variation of diurnal tidal amplitudes, with tidal modulation also present in the annual variation at
mid-latitudes. Gao et al. (2010) used the OH airglow emission data from TIMED/SABER observations to analyze which
oscillations dominate the OH airglow emission and the distribution of these oscillations. Strong semiannual oscillations in the
OH airglow emission in the equatorial region have been observed using different observational methods. The maximum value
of OH airglow emission in the equatorial region occurs at the equinox and the minimum value occurs at the solstice. With
increasing latitude, annual oscillation dominates at higher latitudes.
Not only about the seasonal variation of OH airglow emission but also about the long-term variation of OH airglow emission
has made great progress. Wiens and Weill (1973) analyzed the diurnal, annual, and solar cycle variations of OH airglow by
studying OH airglow intensity data observed by filter photometers in the tropics and at latitude stations in the northern and
southern temperate zones. They found that the diurnal variation pattern is a function of latitude and season, and indicated that
the OH airglow intensity is influenced by solar activity. Batista et al. (1994) found a positive correlation between the OH(9,4)
airglow intensity obtained in Brazil (23° S, 45° W) and the $F_{10.7}$ index by analyzing the relationship between this intensity and
the intensity of solar activity. Liu and Shepherd (2006) analyzed the relationship between OH airglow emission and solar
radiation using the WINDII observation data from 1991 to 1997 and pointed out that OH airglow emission is dependent on





solar radiation. Baker et al. (2007) found that the experimental data from SABER for several years exhibited equatorial
enhancements of the nighttime mesospheric OH airglow layer consistent with the high average diurnal solar flux. Pertsev and
Perminov (2008) analyzed the response of hydroxyl airglow to solar activity using observations from the Zvenigorod
Observatory (56° N, 37° E) and found that the response of emission intensity to the variation of $F_{10.7}$ solar radio flux was 30%-
40%/100sfu, with a more significant response in winter compared to summer. Reid et al. (2014) analyzed hydroxyl (OH)(8-3)
airglow data obtained from filter photometer measurements near Adelaide, Australia. The results show that the OH airglow
intensity is related to solar activity. Many studies have shown that the intensity of OH airglow is closely related to solar activity,
but the emission height of OH airglow shows a different response. Von Savigny (2015) analyzed the vertical volume emission
rate of OH(3-1) airglow observed by the Scanning Imaging Absorption Spectrometer of the Atmosphere (SCIAMACHY) on
the Envisat satellite and found no significant solar cycle signature for the OH emission height. Gao et al. (2016) used the
airglow data observed by SABER to analyze the response of NO, O2, 2.0 μm and 1.6 μm OH airglow to solar radiation, and
they found that the intensity and peak emission rate of the four airglow emissions were strongly correlated with solar radiation,
but the response of NO, 1.6 μm and 2.0 μm OH airglow peak heights to solar radiation was not significant. They also pointed
out that the response of airglow emission to solar radiation varies with latitude.
Numerical models are often analyzed in conjunction with observational data to study the relationship between OH airglow
radiation changes and dynamical processes and solar activity changes. Yee et al. (1997) analyze three-day observations from
the High-Resolution Doppler Imager (HRDI) instrument on UARS and simulations from the Thermosphere-Ionosphere-
Mesosphere-Electrodynamics General Circulation Model (Time-GCM). They show that the vertical motion, driven mainly by
tides, changes the atomic oxygen profile, leading to changes in the observed peak height and brightness of the airglow. The
upward motion reduces the atomic oxygen concentration at the top height of the mesosphere, thus reducing the OH airglow
intensity. Marsh et al. (2006) examined the variations of OH Meinel airglow by comparing SABER measurements of OH
airglow with predictions from a three-dimensional chemical kinetic model. They found that migrating diurnal tides have a
considerable influence on diurnal and seasonal variations at low latitudes, while at high latitudes the annual variations in
emissions are mainly due to the transport of oxygen by the seasonally reversed mean circulation. Huang (2016) used OHCD
and MACD models to simulate OH airglow and O airglow to study the effects of $CO_2$ gas concentration variation and solar
cycle variation on airglow intensity and volume emission rate (VER). The results show that the effect of solar cycle variation
on airglow variation is greater than the effect of $CO_2$ gas concentration variation on airglow variation.

Much progress has been made in the study of the seasonal and long-term variations of the OH airglow and the global
distribution characteristics of the amplitude and phase of the variations, but many details of these variations have not yet been
fully described. In addition to OH airglow intensity, peak emission rate and peak height are important parameters that can be
used to describe the airglow emission rate. OH airglow can provide important information for the study of mesospheric
dynamics and chemistry, and its long-term variation may be modulated by the cyclic variation of solar activity in addition to
the dynamics. Studying the seasonal and annual variation of OH airglow emission in the MLT region and the possible



modulation of OH airglow emission by solar activity can help advance the understanding of the underlying processes of energy,
chemistry, dynamics, and transport in the mesosphere and low thermosphere.
In this paper, we fit the OH airglow emission rates observed by SABER during 2002-2022 to obtain the peak emission rates
and peak heights and use them as parameters to describe OH airglow emissions. The seasonal and interannual variations of
OH airglow peak emission rate and peak height are analyzed. The data and analysis methods are presented in Section 2. Section
3 analyzes the semiannual, annual, and interannual variation of the peak emission rate and the peak height of OH airglow. For
seasonal variations, the SAO and AO of the peak OH airglow emission are analyzed, and the amplitudes and phases of the
SAO and AO are calculated and compared for different latitudes. For interannual variation, the correlation between the QBO
of peak OH airglow emission in the equatorial MLT region and the QBO of stratospheric zonal winds, as well as the influence
of peak emission rate and peak height by solar activity, are discussed, and the possible reasons for this phenomenon are
analyzed. An overview is given in Section 4.
**2 Data and Analysis Method**
**2.1 TIMED/SABER**
The Sounding of the Atmosphere using Broadband Emission Radiometry (SABER) instrument is one of four instruments on
NASA's TIMED (Thermospheric Ionospheric Mesospheric Energy Dynamics) satellite. SABER makes global measurements
of the atmosphere by using a broadband limb-scanning infrared radiometer that measures not only the volumetric mixing ratios
of $O_3$, $CO_2$, H2O, [O], and [H], but also the NO OH emissions in the excited state. The wavelengths of the OH airglow emission
centers measured by SABER are 2.0 μm and 1.6 μ m. The 2.0 μm channel measures the radiation to come from the OH (9, 7)
and OH (8, 6) bands, while the 1.6 μm channel measures the radiation from the OH (5, 3) and OH (4, 2) bands and part of the
OH (3, 1) band.
We use version 2.0 of the 1.6 μm OH data observed by bandpass filters with a central wavelength of 1.6 μm from January 25,
2002, to December 31, 2022. SABER can observe the latitude range from 53° in the winter hemisphere to 83° in the summer
hemisphere. To ensure data continuity at high latitudes and avoid inconsistent analysis due to missing measurements at high
latitudes, we chose to analyze OH airglow emission in the latitude range of 52.5°N-52.5°S. Before analyzing the long-term
variation of OH airglow peak emission rates and heights, we need to extract them from the height profiles of OH airglow
volume emission rates. The peak height of OH airglow emission falls mainly in the range of 80km to 90km, so we use the
Fourier fourth-order fitting method for the observation points near this range:
$V(z) = a_0 + \sum_{i=1}^{4}[a_i \cos(i * \omega * z) + b_i \sin i * \omega * z]$ ,                                                          (1)
where z is the height and V is the OH airglow emission rate, resulting in the peak OH airglow emission rate and peak height,
which are expressed as peak emission rate and peak height, respectively, in the later paper. It is used to carry out the study of
seasonal and interannual variations of peak OH airglow emission. Figure 1 shows an example of the observed and fitted height



profile of the OH airglow volume emission at 48°S on April 1, 2008. The dashed lines in the figure point to the peak emission
rate ($V_{max}$) and peak height ($H_{max}$), respectively.

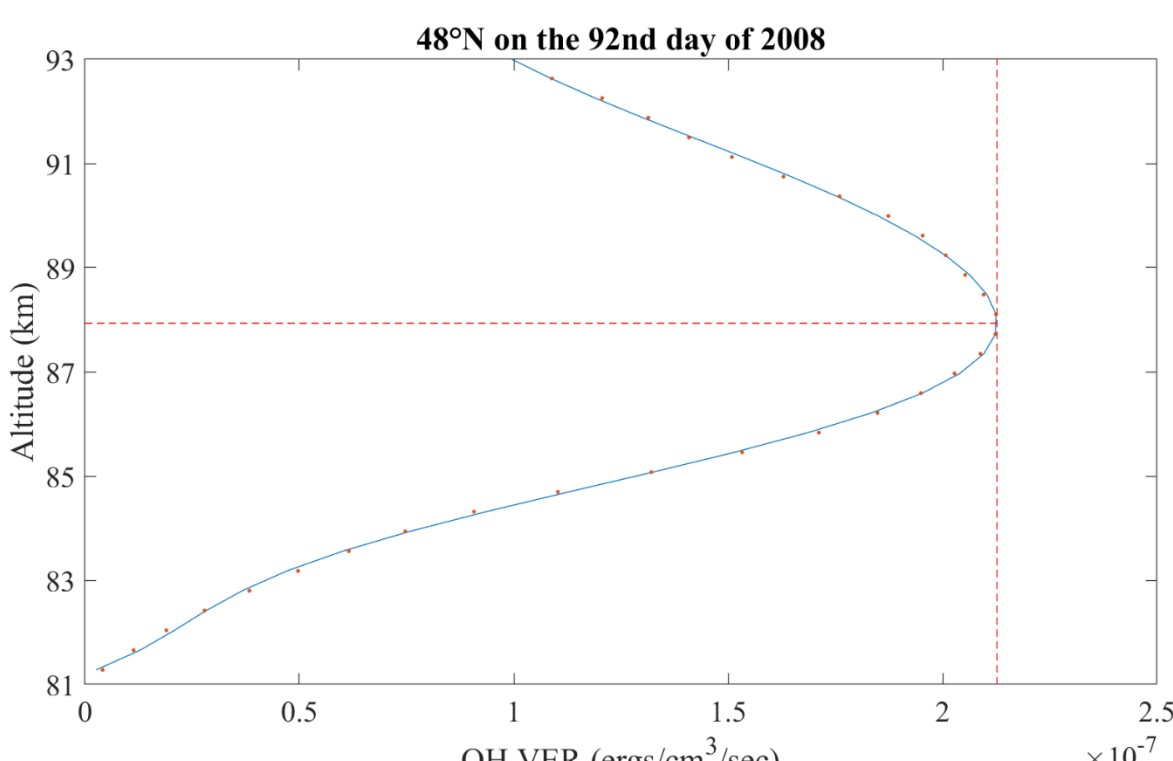


**Figure 1: Observations (points) and fitted curves (solid lines) of OH airglow emission on April 1, 2008, where the units of $V_{max}$ and $H_{max}$ are ergs/cm³/sec and km, respectively.**

## 2.2 LRO

The low-resolution OMNI (LRO) dataset consists mainly of hourly averaged near-Earth solar wind magnetic and plasma
parameters from several spacecraft in geocentric or L1 (Lagrange point) orbit. LRO provides proton fluxes with energies above
1, 2, 4, 10, 30, and 60 MeV from several IMP and GOES spacecraft, and provides a wide range of geomagnetic and solar
activity indices. In this paper, the solar radiation flux $F_{10.7}$ provided by it from 2002 to 2022 is mainly used as an indicator of
the solar activity intensity.



### 2.3 MERRA2

The second Modern-Era Retrospective analysis for Research and Applications (MERRA-2) is a NASA atmospheric reanalysis using version 5.12.4 of the Goddard Earth Observing System Model Version 5 (GEOS-5), which enables the use of newer microwave sounders and hyperspectral infrared radiance instruments, as well as other data types. Unlike MERRA, all data sets for MERRA-2 are available on the same horizontal grid. This grid has 576 points in the longitudinal direction and 361 points in the latitudinal direction, corresponding to a resolution of $0.625° \times 0.5°$. The vertical resolution of the data is changed from $0.667°$ in MERRA and the latitudinal resolution remains the same ($0.5°$). Their wind field data contain up to 80 km of zonal wind data with a temporal resolution of 3 hours. We chose to characterize the equatorial stratospheric QBO using zonal winds at 30km in the Singapore region ($0.625°$N,$103.125°$S).

### 3 Results and Discussion

A grid of $5°$ latitude $\times$ 1 day was constructed to study the long-term variation of OH airglow. The peak emission rate and peak height obtained from the fit were first averaged daily and latitudinally. The averaged results were averaged by latitude range and the result obtained by averaging was used as the value of the grid centroid. In the latitude direction, a total of 21 latitude windows are obtained from $52.5°$S to $52.5°$N. The TIMED satellite needs to operate for about 60 days to cover the global 24h place time with its measurements, and the results after the day averaging are smoothed with a sliding step of 1 day and a window length of 60 days. The smoothed results are shown in Figure 2.



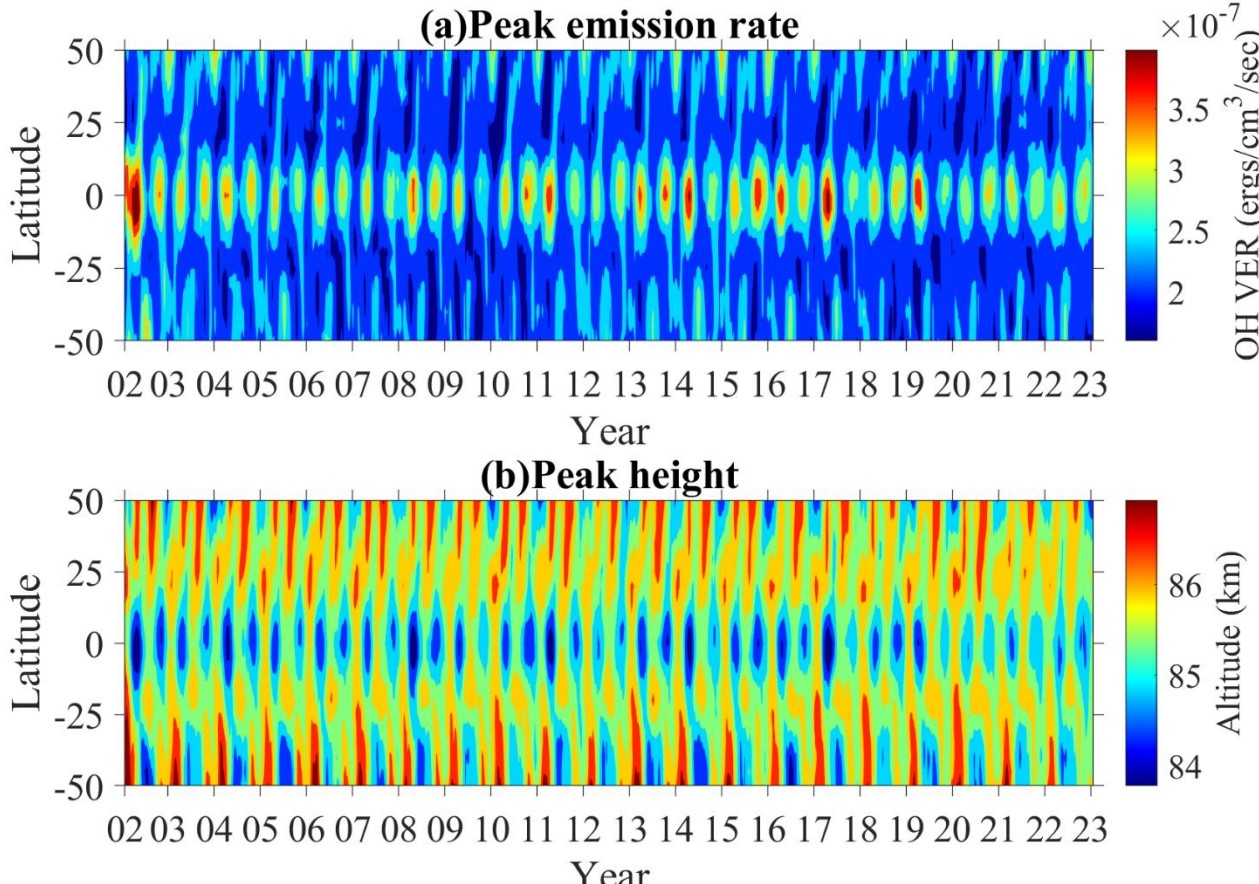

157

**Figure 2: (a)Latitude-time distribution of peak emission rate. (b)Latitude-time distribution of peak height. The units are ergs/cm³/sec and kilometers, respectively. Peak emission rate and peak height are averaged over days and smoothed over a window length of 60 days. The period starts from January 25, 2002, until December 31, 2022.**

**3.1 Annual and Semiannual Oscillation**

In Figure 2, the strongest semiannual oscillations can be observed in the equatorial region, which is consistent with the results observed in the ground experiment of Taylor et al. (2005). To facilitate the study of annual and semiannual oscillation of airglow radiation, airglow radiation data for a total of 21 years from 2002 to 2022 are averaged by superimposing them. A superimposed year is obtained that can be used to represent a multi-year average of peak emission rate and peak height. Figure 3(a) shows the peak OH airglow emission rate after multi-year averaging, and it can be seen that there is a clear semiannual oscillation in the equatorial region. The maximum value is at the equinox and the minimum value is at the solstice, and the extreme value at the September equinox is larger than that at the March equinox, which is consistent with the previous studies (Mulligan et al., 1995; Takahashi et al., 1995; Buriti et al., 2004).

Figure 3(b) shows the OH airglow peak height after multi-year averaging, and the same as the peak emission rate, the peak height has the same semiannual oscillation phenomenon, and the minimal value appears at the equinoxes. OH airglow peak





height is inversely proportional to the peak emission rate. In the equatorial region, the minimum value of the peak height at the equinox corresponds to the maximum value of the peak emission rate. The same phenomenon has been identified in previous studies and is thought to be the role of tides in this (Yee et al., 1997; Melo et al., 1999).

**Figure 3: A superimposed year of peak OH airglow emission. (a) Latitude-time distribution of the peak emission rate averaged over multiple years. (b) Latitude-time distribution of the peak height averaged over multiple years.**

In order to obtain the amplitudes and phases of SAO and AO in the peak OH airglow emission, a harmonic fitting method was used, applying the following Eq. (2):

$$f = f_0 + a * \cos\left(\frac{2\pi}{183(day)}(t - t_{SAO})\right) + b * \cos\left(\frac{2\pi}{365(day)}(t - t_{AO})\right), \qquad (2)$$





where f can be either the peak emission rate or the peak height, $f_0$ denotes the multi-year average, t is the time index in days,
and a and b are the amplitudes of SAO and AO. $t_{SAO}$ and $t_{AO}$ are the times at which the semi-annual and annual oscillation
maxima occur. Figure 4 shows the amplitudes and phases of the annual and semiannual oscillation of OH airglow peak
emission rate and peak height at different latitudes, respectively.
Figures 4 (a) and (b) show the latitudinal variation of peak emission rate relative amplitude in percentage terms: $(a /f_0) \times 100\%$,
$(b /f_0) \times 100\%$. As seen in Figure 4(a) SAO amplitudes range from 2% to 20% with three peaks, with the largest amplitude of
about 20% in the equatorial region. The phase of SAO is delayed from near day 90 (near the equinox) at the equator and then
with increasing latitude to near the solstice at 50°N/S. The peak emission rate has its largest amplitude during the equinox,
which is consistent with diurnal tides (Burrage et al., 1995). The peak emission rate is associated with diurnal tides and its
seasonality is likely to be caused by the seasonal variation of diurnal tides. Notably, the OH airglow peak emission rate in
Figure. 2 is anomalous in 2015, with the maximum value at the March equinox being smaller than the maximum value at the
September equinox, the reason for this occurrence we analyze later.
As shown in Figure 4(b), the amplitude of AO ranges between 1% and 16% with two weak peaks at 5°S and 25°N. There are
three troughs at 30°S, 15°N, and 35°N, respectively. In the latitude range of 30°-50°, the AO amplitude increases with
increasing latitude. The largest amplitude is found at 50°N/S, where the amplitude reaches 16% at 50°N and is greater than
11% at 50°S. Observing the AO phase, the amplitude of 25°N is greatest on day 183 of the first year, delayed towards the
poles. The amplitude of the 50°S reaches its maximum on day 140 of the second year and the 50°N reaches its maximum on
day 337 of the first year. Comparing the amplitudes of SAO and AO, we find that semiannual oscillation dominates at low
latitudes and annual oscillation is more frequent at higher latitudes.
We note the semiannual and annual variations in OH airglow intensity provided by Reid et al. (2014), who analyzed filter
photometer measurements at Buckland Park (34.6°S,138.6°E). They found a value of 7% for SAO, which peaked at day 160
and was in good agreement with our SAO results. The AO is a poorer match, with Reid's amplitude of 14%, several times that
of our results. The phase shows a maximum amplitude at day 159, in general agreement with our results.
The amplitude of the peak height is the absolute amplitude rather than the relative amplitude, as shown in Figure 4(c), and the
trend of the phase and amplitude change of the peak height SAO is the same as that of the peak emission rate SAO. The SAO
amplitude is between 0.3 and 0.6 km. It has three peaks at 40°S, 0° and around 50°N, the equatorial peak being the strongest.
The AO amplitude is below 1km, and the two weak peaks at 5°S and 20°N are significantly lower than those at higher latitudes.
At low latitudes, the strongest variation in peak height is SAO; near 50°, AO is the strongest. The phase of SAO in peak height
is delayed from near the solstice at the equator to near the equinox at 50°S (from the winter solstice to the following equinox).
Observing the AO phase, the amplitude of 20°S is greatest on day 186 of the first year, delayed towards the poles. The
amplitude of 50°S reaches its maximum amplitude on day 2 of the second year and the 50°N reaches its maximum on day 178
of the second year. The amplitude and phase variations of SAO are consistent with the results of Gao et al. (2010), but the
results of the AO differ from ours.

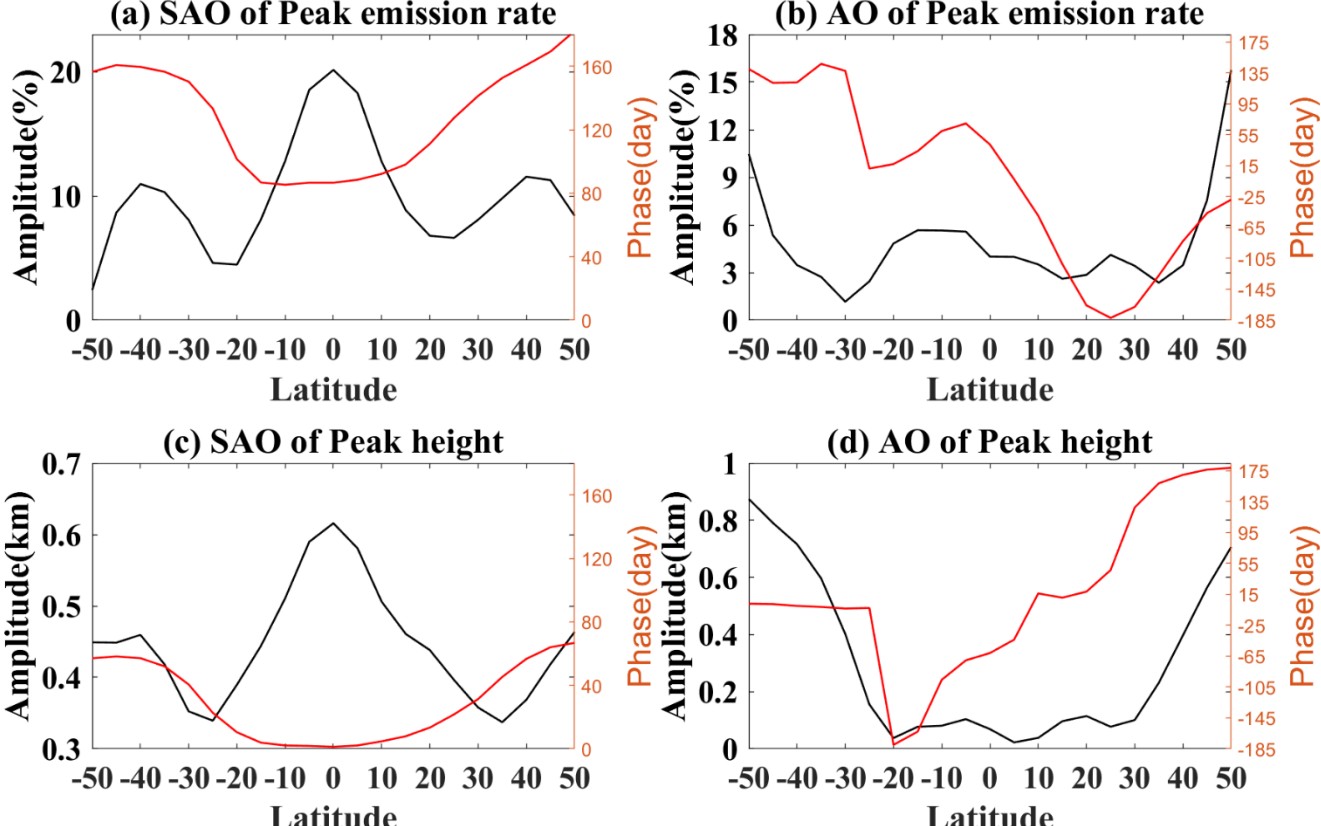

**Figure 4: (a) Latitudinal distribution of amplitude and phase of SAO in the peak emission rate. (b) Latitudinal distribution of amplitude and phase of AO in the peak emission rate. (c) Latitudinal distribution of amplitude and phase of SAO in the peak height. (d) Latitudinal distribution of amplitude and phase of AO in the peak height.**

**3.2 Modulation by QBO**

The production of hydroxyl groups (OH *) is associated with the following reactions: $O_3 + H \rightarrow OH^* + O_2$. As a reaction product, the rate of OH* production is directly related to $[O_3]$ and $[H]$ and the rate of reaction depends on the local temperature and density. In addition, $[O_3]$ is proportional to $[O]$ under photochemical equilibrium conditions. Thus, OH emission is also related to $[O]$. The atomic oxygen distribution derived by Russell and Lowe (2003) using the hydroxyl emission and oxygen green line emission observed by WINDII exploits precisely the relationship between OH emission and atomic oxygen.

At low latitudes where SAO dominates the hydroxyl airglow emission, Burrage et al. (1996) analyzed horizontal wind field measurements obtained from 1992 to 1995 by the High-Resolution Doppler Imager experiment on UARS and found an oscillation with a period of about 2 years, with the maximum amplitude occurring near an altitude of 85 km, which corresponds to the peak height in our results. One of the mesospheric quasi-biennial oscillation (MQBO) shows a similar spatial distribution structure to the mesospheric semiannual oscillation (MSAO) and exhibits a phase relationship with the stratospheric quasi-





two-year annual oscillation, suggesting that the MSAO is modulated by the stratospheric quasi-two-year oscillation. In addition,
the Christmas Island MF radar (2°N, 130°W) also detects an MQBO with the same phase and the same peak height, although
the amplitude is only half of that shown by HRDI, this observation also confirms the existence of this MQBO.
To facilitate the analysis of the relationship between OH airglow emission and stratospheric QBO, we smoothed peak emission
rate and peak height with a window of 365 days in length, which avoids the effect of seasonal variations on the analysis of the
results.

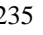


fast



**Figure 5: (a)Latitude-time distribution of peak emission rates in ergs/cm³/sec. Emission rates are smoothed over a window length of 365 days. The white line refers to the daily average of the equatorial stratospheric zonal winds, again smoothed over a window length of 365 days. Eastward winds are indicated by positive values. (b) Same as (a) but for peak height.**

Figure 5(a) shows the latitude-time distribution of peak OH airglow emission rates, with the white line being the daily average of the equatorial stratospheric zonal winds. After 365 days of smoothing, it can be observed that in the equatorial region, the QBO signal in the OH airglow emission may be related to the QBO signal in the stratospheric latitudinal winds. The peak emission rate QBO signal is much weaker in the higher latitude region than in the equatorial region, which is also consistent with previous findings (Marsh et al., 2006; Shepherd et al., 2006). Observations of the phase of the peak emission rate in the equatorial region show that the phase change of the emission rate is consistent with the phase change of the tropical zonal wind at 30 km. The OH airglow emission rate is relatively high when the zonal winds are to the east, as in 2011. When the zonal winds are westward, the OH airglow emission rate is relatively low, e.g. in 2010. It is worth noting that the anomalies in OH airglow emission in 2015 mentioned above are associated with stratospheric QBO anomalies. The QBO anomalies in OH airglow emission rate from 2015-2016 correspond to the anomalous changes in QBO observed by radio soundings by Newman et al. (2016), which also confirms that OH airglow emission at 85 km is indeed affected by QBO in the tropical stratosphere influence.

Figure 5(b) shows the latitude-time distribution of peak OH airglow heights, with the white line being the daily average of the equatorial stratospheric zonal winds. In the equatorial region, the peak height is modulated by the QBO as is the peak emission rate. Of course, since peak height is inversely proportional to peak emission rate, a high peak emission rate corresponds to a low peak height. The QBO signal for peak height is also weak at higher latitudes. Peak heights show more short-period variation, while peak emission rates show long-period variation, e.g. the peak emission rates in 2002 and 2014 are significantly larger than that in 2008 and 2019, which we will analyze later.

The QBO signal in the equatorial region has been the subject of many related studies. Xu et al. (2009) analyzed the quasi-biennial oscillation of the migrating diurnal tide based on data from TIMED observations. After comparison, we find that the distribution of OH airglow emission at low latitudes is similar to that of the migrating diurnal tide in temperature. The QBO phenomenon is more pronounced in equatorial regions compared to higher latitudes, due to the dominant tidal influence on OH airglow emission at lower latitudes. Pramitha et al. (2021) found a good correspondence between the mesosphere-low thermosphere diurnal tide and the stratospheric quasi-biennial oscillation (SQBO) based on meteor radar observations and Whole Atmosphere Community Climate Model (WACCM) simulations. As to how stratospheric QBO modulates the emission of OH airglow, Shepherd et al. (2006) found a decrease in the center of the airglow emission maximum during QBO modulation, indicating the downward transport of atomic oxygen. This is also verified by the significant decrease in peak height in Figure 5(b) when the zonal wind is eastward compared to when the zonal wind is westward. They suggest that the tidal influence on the mesospheric airglow emission is due to a corresponding tidal change in the atomic oxygen mixing ratio caused by vertical motion. The stratospheric QBO phase variation is consistent with the tides in the MLT region and the airglow emission phase variation. The QBO likely modulates the airglow emission at low latitudes through the tides.





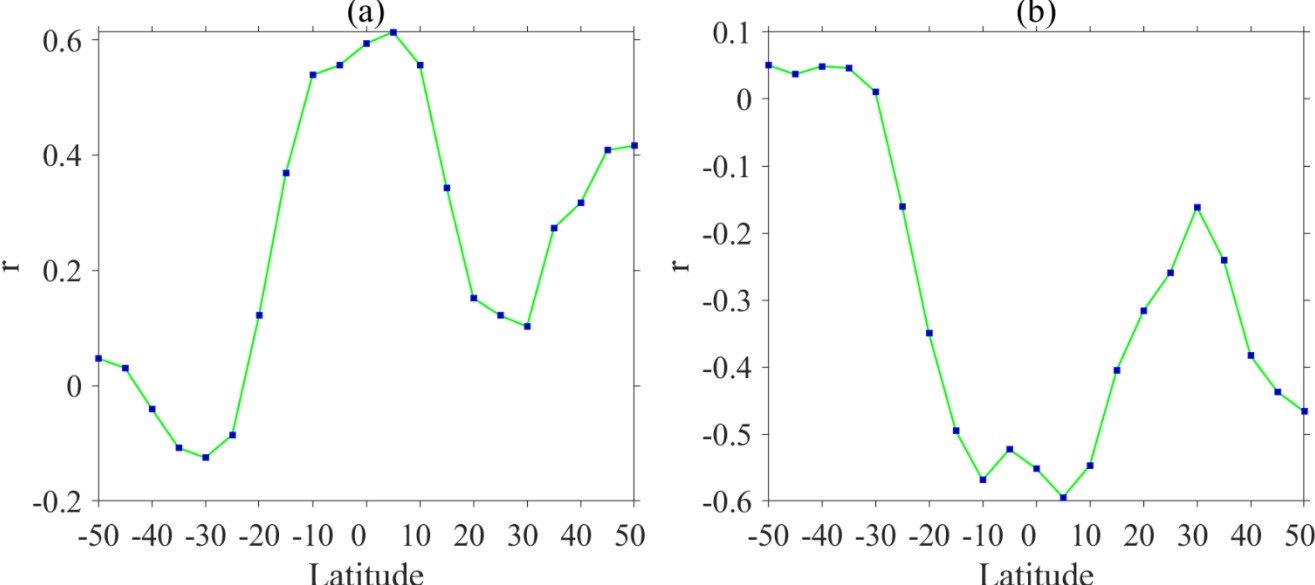

**Figure 6: (a) Latitudinal dependence of the correlation coefficient between peak emission rate and QBO in the stratosphere; (b) Latitudinal dependence of the correlation coefficient between peak height and QBO in the stratosphere.**

To investigate the relationship between the peak OH airglow emission and the QBO of the tropical stratospheric atmosphere at different latitudes, we calculated the correlation coefficient between OH airglow and stratospheric QBO. Figure 6 shows the correlation with latitude, where Figure (a) shows the correlation between peak emission rate and stratospheric QBO, with the strongest correlation in the equatorial region with a correlation coefficient of 0.6. The comparison shows that the correlation is not the same in the two hemispheres. In the southern hemisphere, the correlation begins to weaken with increasing latitude, with a correlation coefficient of 0.12 at 20°S, and then becomes negative, with a correlation coefficient of -0.12 at 30°S. In the northern hemisphere, the correlation coefficient decreases with increasing latitude, with the smallest correlation coefficient of 0.1 at 30°N. The correlation coefficient then increases with increasing latitude, with a correlation coefficient of 0.42 at 50°N. Figure (b) shows the correlation between the peak height and stratospheric QBO, similar to the peak emission rate in (a), and again the two are most correlated at the equator with a maximum correlation coefficient of -0.59. Comparing the two hemispheres, in the southern hemisphere the correlation is negative at latitudes greater than 30°. In the northern hemisphere, the absolute value of the correlation coefficient increases at latitudes greater than 30°. Comparing the different latitudes, we can find that the OH airglow emission responds most strongly to QBO in the equatorial region and is modulated by stratospheric QBO. As latitude increases, the QBO signal of OH airglow emission weakens. The modulation by QBO differs between the two hemispheres, with the correlation coefficient weaker in the southern hemisphere than in the northern hemisphere at the same latitude, especially at latitudes greater than 30°.



### 3.3 Modulation by Solar Activity

In addition to seasonal variations, at certain latitudes, e.g. around 50° latitude, the OH airglow appears to have a greater peak emission rate during 2002-2003 and 2014-2015 than near 2008-2009 and 2019-2020. Airglow is very sensitive to atmospheric conditions, and solar variations also affect atmospheric conditions such as temperature and gas concentration, with any change in temperature or gas concentration leading to a change in airglow intensity. Since solar radiation directly drives the production of atomic oxygen, then OH airglow emission is likely to vary with solar activity, i.e. it is modulated by solar activity. Scheer et al. (2005) analyzed rotational temperature and airglow brightness variations in the OH (6-2) and O-2 (0-1) bands measured at El Leoncito (32°S, 69°W) from 1998 to 2002 and found that at 87 km, there was no correlation with the solar radio flux, while a very strong correlation developed at 95 km. This section investigates the correlation between peak global OH airglow emission and solar activity by using SABER-sounding data and the 10.7 cm solar radiation stream ($F_{10.7}$) dataset.

On top of smoothing with a 365-day-long window, a three-year window was used and the smoothed results are shown in Figure 7(a) and Figure 7(b). $F_{10.7}$ data were smoothed in the same way, and to ensure consistency and accuracy of the smoothed results, the first and last two years of data were removed, and we chose the data for the time range 2004 to 2020. Where (a) indicates the latitude-time distribution of peak emission rate and (b) indicates the latitude-time distribution of peak height. As can be seen from the results shown in Fig. 7(a), OH airglow emission rates are influenced by solar activity, with peak emission rates significantly greater in high solar activity years such as 2014 than in low solar activity years such as 2008 and 2019. Notably, the response of peak emission rate to solar activity in equatorial regions differs from that at higher latitudes, with an advance in the response of OH airglow emission in 2008, a low solar activity year. The same phenomenon was found by Clemesha et al. (2005) when they studied the monthly mean OH(6-2) band intensity at Cachoeira Paulista (23°S, 45°W) during the period 1987 to 2000. They found that the maximum intensity occurred about 1.4 years before the maximum of solar cycle 22. Deutsch and Hernandez (2003) also found that the response of OI 558 nm to solar activity shows some lag, possibly because the decline in emission intensity after the solar maximum is slower than the decline in solar activity. The peak height in Fig. 7(b) also shows some response to solar activity but is significantly weaker than the response of the peak emission rate. Von Savigny (2015) found no clear long-term trends or 11-year solar cycle features in the OH emission height time series.

Figure 7(c) shows the annual average change in OH airglow peak emission rate (black dot and line) over the period 2004-2020, calculated from smoothed data. The trend in the global annual average of the peak emission rate closely matches the change in $F_{10}$.7 (red star and line) shown in Fig. 7c, with a correlation coefficient of 0.89. These results suggest that the change in peak OH airglow emission may be modulated by solar activity. Figure 7(d) shows the global annual mean variation in peak height, from which we can see that the peak height, although showing an opposite trend to solar activity, has a correlation coefficient of only -0.66, and the peak height is probably less influenced by solar activity.



**Figure 7: The interannual variations of peak OH airglow emission. (a) Latitude-time distribution of peak emission rates in ergs/cm³/sec. Emission rates are smoothed over a window length of 3 years. (b) Latitude-time distribution of peak height in km. (c) The global year average of peak emission rate and $F_{10.7}$ during 2004-2020. (d) The global year average of peak height and $F_{10.7}$ during**





**2004-2020. (e) The scatterplot of $F_{10.7}$, peak emission rate, and the solar response value of peak emission rate. (f) The scatterplot of**
**$F_{10.7}$, peak height, and the solar response value of peak height.**
Analysis of the solar cycle dependence on the peak emission rate and its solar response. A global annual mean series scatter
plot between the OH airglow peak emission rate and $F_{10.7}$ is shown in Fig. 7(e), and that of the peak height is shown in Fig.
7(f). From Fig.7(e) and Fig. 7(f), we find a clear linear relationship between the OH airglow emission and the corresponding
annual average of $F_{10.7}$. Assuming a linear relationship, we use linear regression to calculate the solar response to OH airglow.
To investigate the solar activity dependence of the OH airglow. Using a linear approximate regression model for describing
the relationship between OH airglow and solar radiation flux $F_{10.7}$:
$\qquad f = A + B * F10.7$ , $\qquad\qquad\qquad\qquad\qquad\qquad$ (3)
where B is the slope of the linear regression equation and A is its ordinate at the origin; that is, A is a constant and B is the
coefficient of linear fit derived from the least-squares regression analysis, representing the solar response to the OH peak
airglow emission rate. Using least-squares regression calculations, we obtain the global response of the OH peak airglow
emission rate to solar activity and its standard deviation. The solar response value for the peak emission rate is $(1.91 \pm 0.24)$
$*10^{-8}$ ergs/cm$^3$/sec/100 sfu as shown in Figure 7(e). The solar response value for the peak height is $(-0.07 \pm 0.02)$ km/100 sfu
as shown in Figure 7(f).
After analyzing the response of the global (50°S-50°N) mean of OH airglow emission to solar activity, the response of different
latitude zones to solar activity is analyzed below. A correlation analysis of the annual mean series for each latitudinal zone
yielded correlation coefficients between the peak emission and $F_{10.7}$ for 21 latitudinal zones, and the results are shown in Fig.
8(a). The correlation coefficients for the higher latitude zones are all greater than 0.6, except for the equatorial zone where the
correlation coefficient is only 0.42. The largest correlation coefficient is 0.96 for 30°S. Figure 8(b) shows the correlation
coefficients between peak height and $F_{10.7}$ for each latitude zone. The correlation coefficients are less than 0.6 for all latitude
zones except near 30°S where the correlation coefficient is greater than 0.6.



**Figure 8: The latitudinal dependence in the solar response of peak OH airglow emission. (a) The correlation coefficients between peak emission rate and $F_{10.7}$ for 21 latitude zones. (b) The correlation coefficients between peak height and $F_{10.7}$ for 21 latitude zones. (c) The solar response ranges of peak emission rate for 21 latitude zones, and the vertical bars show the standard deviation of solar response. (d) The solar response ranges of peak height for 21 latitude zones, and the vertical bars show the standard deviation of solar response.**

Using the least squares method, we obtained the latitudinal distribution of the peak emission rate response to solar activity and its standard deviation. Figure 8(c) shows the magnitude of the peak emission rate response to solar activity and its standard deviation for each latitude zone. The response amplitude is smallest at the equator and increases with increasing latitude. The solar response values range from a minimum of $0.14*10^{-7}$ ergs/cm$^3$/sec/100 sfu at the equator to a maximum of $0.24*10^{-7}$





ergs/cm$^3$/sec/100 sfu in the 50°S latitude zone. Figure 8(d) shows the magnitude of the peak height response to solar activity
at each latitude zone and its standard deviation. The response is weak at the equator and in the northern hemisphere, with the
strongest response near 25° in the southern hemisphere at -0.13 km/sfu. A comparison of the latitudinal distribution
characteristics of the solar response to OH airglow emission reveals that the solar response in the southern hemisphere is
greater than that at the corresponding latitude in the northern hemisphere. In each hemisphere, the standard deviation of the
solar response increases gradually from 30° to the equator and the poles. Tang et al. (2018), in using the SABER and $F_{10.7}$
datasets to study the correlation between peak emission rate and solar activity, found that peak emission rate was significantly
correlated with the solar cycle and that the correlation coefficient was also higher in the Southern Hemisphere than in the
corresponding latitudes in the Northern Hemisphere. In addition, the effects of planetary waves, tides, and the coupling between
them on airglow may vary with latitude and solar activity. A study by Laskar et al. (2013) showed that during periods of low
solar activity, the influence of the lower atmosphere on the upper atmosphere was found to be stronger. During periods of high
solar activity, there is a clear correlation between the behavior of the upper atmosphere and direct solar forcing, but the effect
of lower atmospheric wave activity on the upper atmosphere is weaker. Laskar et al. (2014) found that the effect of planetary
waves and the coupling of planetary waves and tides on the upper atmosphere varies with solar activity. These may also
contribute to the latitudinal dependence of the airglow response to solar activity.
Figure 9 illustrates the mechanism by which QBO and solar activity regulate OH airglow. Marsh et al. (2006) compared model
and observational data and found that most of the variability in OH airglow emission is caused by changes in the production
rate of ozone. At the height of the peak emission, the variation is mainly caused by changes in atomic oxygen due to vertical
transport. For QBO, we find that the phase variation of OH airglow emission in the equatorial region remains consistent with
the stratospheric QBO phase variation. Dynamic processes dominate the effect on the emission rate, and the increase in
emission rate is mainly caused by the increase in the mixing rate of atomic oxygen produced by downward tidal motion. As
the stratospheric zonal wind field moves eastward, the enhanced downward motion coincides with enhanced atomic oxygen
concentrations, enhanced OH airglow emission rate, and minimization of OH airglow emission height. We, therefore, speculate
that tides play an important role in the modulation of OH airglow emission by QBO. In terms of solar activity, the production
of atomic oxygen is influenced by solar radiation, so the OH airglow emission rate exhibits variations in the solar cycle. In
contrast, the variation in peak airglow height is mainly driven by tidal-driven vertical motion that changes the distribution of
atomic oxygen, so the variation in peak airglow height is less influenced by solar activity.





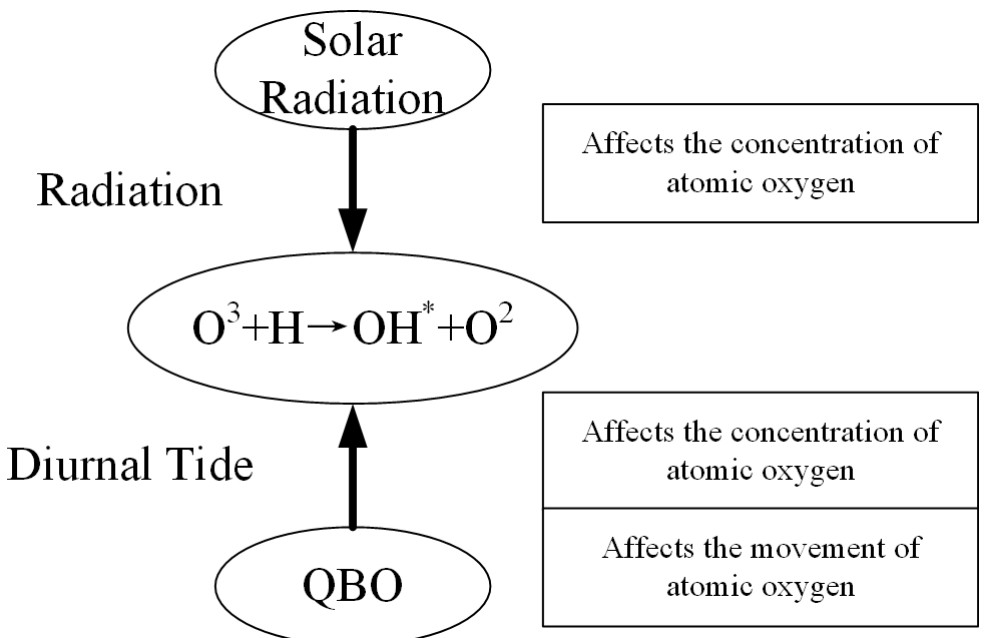


**Figure 9: Schematic representation of the mechanism by which OH airglow is modulated by QBO and solar activity.**


**4 Summary**
In this paper, based on the observed OH airglow emission data from the TIMED/SABER satellite from 2002-2022, the peak
emission rate and a corresponding height of OH airglow were obtained by Fourier fitting method, and the seasonal and
interannual variations of the peak emission rate and peak height were analyzed. For the seasonal variation, the peak emission
rate and height were superimposed and averaged to observe the semiannual oscillation and annual oscillation of the peak
emission rate and height. As a result, the latitudinal variations of peak emission rate and height were similar. The SAO
amplitude ranges between 2% and 20% for peak emission rate, with the largest amplitude of about 20% in the equatorial region
and two peaks larger than 40° N/S. The AO amplitude ranges between 1% and 16%, with two weak peaks at 10° S and 25° N,
with the weakest peak at 25° N. The maximum value of the SAO amplitude occurs during the equinox, which is consistent
with diurnal tides and verifies that the OH airglow emission is influenced by diurnal tides. The phase of the SAO is delayed
from near the equinox at the equator to near the solstice at 50°S / N. For the phase of the AO in peak emission rate, the
maximum amplitude first appears on day 183 of the first year at 25°N, delayed towards the poles. The maximum amplitude
occurs at 50°S on day 140 of the second year and at 50°N on day 337 of the first year. For peak height, the SAO amplitude
has three peaks, at 40° S, 0° and around 50° N, with the strongest peak at the equator. The AO amplitude has two weak peaks
at lower latitudes at 5° S and 20° N, which are smaller than those at higher latitudes. The phase of the SAO at peak height is
delayed from near the solstice at the equator to near the equinox at 50° N/S. For the phase of the AO in peak height, the AO is
delayed from 20°S on day 186 of the first year to day 2 of the second year at 50°S and day 178 of the second year at 50°N.



Semiannual oscillations dominate at lower latitudes and, with increasing latitudes, annual oscillations dominate at higher latitudes.

The OH airglow emission also shows a QBO signal for interannual variations, especially in the equatorial region. The peak emission rate and height are smoothed so that the phase variation remains consistent with the stratospheric zonal winds. When the wind field is to the east, the peak emission rate is relatively large and the peak height is relatively low. Comparing different latitudes, we find that the correlation between the peak OH airglow emission and stratospheric QBO is strong in the equatorial region, which is modulated by QBO, while the QBO signal is weak in other latitudes. OH airglow emission variation is influenced by changes in atmospheric temperature and related atmospheric components, and as an important dynamic process affecting OH airglow emission variation, migrating diurnal tides also have QBO. QBO in the stratosphere, QBO in the tide, and QBO in the OH airglow emission have similar phase changes. Since the tidal influence on mesospheric airglow emission is through the corresponding tidal change in the atomic oxygen mixing ratio caused by vertical motion, QBO likely modulates airglow emission at low latitudes through tides.

In addition, the correlation between peak OH airglow emission and solar activity has been analyzed. The observed changes in peak OH airglow emission rate correlate well with changes in solar activity over the period 2004-2020, with a correlation coefficient of 0.89, while peak OH airglow emission heights show no significant solar cycle variations. We believe this may be because the peak emission rate will be larger in years of high solar activity. After all, solar radiation affects the production of atomic oxygen. However, the variation in peak airglow height is mainly driven by tidally driven vertical motions that alter the distribution of atomic oxygen, and therefore the variation in peak airglow height is less influenced by solar activity. The solar response of the global peak OH airglow emission is $(1.91 \pm 0.24) *10^{-8}$ ergs/cm$^3$/sec/100 sfu. The latitudinal distribution of peak emission rate and its correlation with the solar cycle is then presented for 21 latitudinal regions. Latitudinal correlation analysis shows that the peak OH airglow emission is significantly correlated with the solar cycle at every latitude from 50°S to 50°N, except for the equatorial region. The solar response distribution of the peak OH airglow emission shows a strong south-north asymmetry between the two hemispheres, with the solar response in the southern hemisphere being higher than that of the corresponding latitude in the northern hemisphere. The solar response is weakest in the equatorial region. The standard deviation of the solar response in each hemisphere increases gradually from 30° to the equator and the poles.

*Code availability.* The code is available at http://www.doi.org/10.57760/sciencedb.space.00659.

*Data availability.* The OH airglow data can be accessed via https://saber.gats-inc.com. The MERRA-2 data (MERRA2_300.tavg_) can be accessed via http://disc.gsfc.nasa.gov. The $F_{10.7}$ data can be accessed via https://omniweb.gsfc.nasa.gov/form/dx1.html.



*Author contributions.* DW processed and analyzed the data and wrote the manuscript; SYG, YFW, and LT reviewed and edited
the manuscript.

*Competing interests.* The authors declare that they have no conflict of interest.

*Acknowledgements.* This work was performed in the framework of space physics research (SPR). We are very grateful to the
SABER team for providing the airglow data. We thank NASA for the MERRA-2 wind field data and $F_{10.7}$ data.

*Financial support.* This research has been supported by the National Natural Science Foundation of China(grant nos. 41831071
and 42188101).

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
