# Peer review of "SAO, AO, QBO, and Long-term trend of the peak OH airglow emission"

_EGUsphere, 2023_

## Author Comment (AC1)

We thank the reviewers and editors for their constructive comments on our manuscript. The manuscript is revised thoroughly by considering all the comments. Our responses to every comment are listed below with blue.

**Response to Anonymous Referee 1**

**Summary**

This paper presents an analysis of the peak altitude and peak emission rate of the hydroxyl (OH) emission rates observed for the past 20 years by the SABER instrument on the NASA TIMED satellite. SABER measures OH emission in two distinct spectral intervals, one centered at 2.0 um and the other at 1.6 um. The paper analyzes and presents results for the 1.6 um channel. The 20 years of SABER data enable comprehensive analysis of various temporal features that appear in the peak height and peak altitude of the OH emission. The analysis presented in the paper shows clear evidence for semi-annual and annual oscillations in the peak features and evidence for influence of the stratospheric quasi-biennial oscillation on the peak features. The results presented often confirm the results of previous papers regarding the presence of temporal features in the OH emission. The paper is clearly written.

The results in the paper are presented as 'engineering' analysis of the temporal variations in the OH emission. There is very little, if any, quantitative physics or chemistry given to explain the observed behavior. As an example, the discussion of solar variability is primarily of correlations between the OH variability and the F10.7 solar radio flux index. Solar variability affects temperature, composition, and dynamics. But the paper does not attempt to quantify which of these effects is dominant. The paper does not explain why there is strong latitude dependence in many of the analyses. However, the SABER dataset includes temperature, atomic oxygen, atomic hydrogen. These datasets could be explored along with model simulations (the WACCM model would be ideal for this) to put the results in context. In addition, although a secondary concern, the paper does not state why the peak altitude and peak emission rates are important physically. They are clearly markers for atmospheric variability. But is the variability important and why? One could imagine that since the reaction of H + O3 is the largest source of heating in the mesopause region, the variation in intensity and location of the OH emission means that the energetics of the mesopause are being altered significantly. Discussions such as this are necessary to place the results in a physical context as the reported variability largely reproduces prior works.

**Recommendation**

The recommendation is to reject the paper and invite submission of a new paper that contains much more detailed physical explanation of the observed OH behavior. This should include comparisons with the WACCM model and evaluations with other SABER data products. The paper should provide a quantitative explanation of features such as the latitudinal dependence of the semiannual and annual oscillations and should go into detail about the relative roles of temperature, chemistry, and dynamics in producing the observed variability.

We have completely revised the manuscript. The physical interpretation of the observed OH

emission variations is addressed in the responses to specific comments below. In addition, we have also used also the WACCCM model data and other SABER data products to provide a quantitative interpretation of features such as the latitudinal dependence of the semiannual and annual oscillations, and to detail the relative roles of temperature, chemistry, and dynamics in producing the observed variability.

A second recommendation is to evaluate the variability of the OH emissions on pressure surfaces and not on altitude. Studying variations at fixed altitudes mix variations in emission as the pressure surfaces rise and fall around the altitudes as the atmosphere warms and cools over the year and over the solar cycle. Pressure is the natural vertical coordinate of the SABER data.

While pressure is the natural vertical coordinate of the SABER data, the SABER data also provide altitude. One of the aspects we study is the variation in OH emission heights, and we do not study variations at a fixed altitude, but rather variable heights, both heights corresponding to peak emission rate and centroid altitude. The centroid altitude, i.e. altitude weighted by the emission rate profile.

**Specific Comments**

Title – the title contains the word 'trend' in relation to the peak OH emission. The word 'trend' typically implies looking at the long-term change of a parameter due to some forcing that is fundamentally changing the atmosphere such as increasing carbon dioxide concentrations. The original time series is analyzed in a way to remove variability (such as the AO, SAO, QBO, and solar cycle) and the linear trend of the residual is computed to derive a change (typically parameter per decade units). The paper does not appear to contain a trend analysis of this type. Please correct the title and the few places in the text where the word 'trend' occurs

We have changed the title to SAO, AO, QBO, and Long-term variation of the OH airglow emission. The word "trend" has also been corrected where it appears in the text.

Data use – An error was discovered in the SABER data for dates after December 16, 2019. A new version, v2.08 is available for data after that date. Please visit the SABER data web site to review and please discard all v2.0 data after Dec 16 2019.

We discard all v2.0 data from 15 December 2019 onwards. Data from 15 December 2019 to 31 December 2022 use the new version v2.08.

Line 23-24. This sentence is an example of the lack of quantitative understanding of the airglow that comes across in the paper. The airglow intensity is not directly related to density and only indirectly related to temperature through the temperature dependence of the rate coefficient for the reaction of H and O3. The entire Introduction is full of generalities that makes one question whether the paper truly understands the physics of airglow generation including how and why it varies. A revised Introduction should directly address the physics/chemistry of OH formation and how it may vary, thus setting up the results and analysis with a model such as WACCM later in the paper.

The energy required for the airglow emission is supplied by solar electromagnetic radiation.

Through direct and indirect action, solar radiation excites atoms, molecules and ions in the upper atmosphere to higher energy states, and activates particles to emit photons when they transition from high energy states to lower energy states, i.e., generating airglow. The hydroxyl (OH) vibration-rotational band is generated by the radiation transition between the vibrational energy levels of the ground state of the OH electron. The source of its production mainly comes from the chemical reaction between hydrogen atoms and ozone:

$$O_3 + \text{H} \xrightarrow{k1} OH(v \leq 9) + O_2$$

Line 116-117 – please provide a reference citation to the vibrational states that contribute to each of the SABER OH channels.

Russell, J., Mlynczak, M., Gordley, L., Tansock, J., and Esplin, R.: Overview of the SABER experiment and preliminary calibration results, SPIE's International Symposium on Optical Science, Engineering, and Instrumentation, SPIE1999. It's covered in this paper.

Line 125 – the instantaneous field of view of the SABER instrument is 2 km. SABER samples the atmosphere at a much higher cadence than every 2 km and so it may appear that the vertical resolution is much higher. The paper needs to discuss the effects of the finite field of view on the ability to determine and analyze variations of the peak height and emission.

The vertical resolution of the SABER observations is about 0.4 km, and the peak emission rate and peak height we obtained using a fit. The fitting corresponds to an altitude interval of 80-100 km, which has enough observation points to satisfy our need to determine and analyze the peak emission rate and peak altitude variations.

Line 136 – please provide a reference citation to the OMNI database and spell out the acronym.

For solar activity, we have used the solar irradiance representation measured by SEE (The Solar EUV Experiment). The F10.7 provided by OMNI, which was used previously, does not have the acronym found at URL https://omniweb.gsfc.nasa.gov/html/ow_data.html, where the data are provided.

Line 161, Section 3.1. The authors are encouraged to examine the data with Fourier techniques to see if other periodic features are evident.

[Figure]

Figure 1: Lomb-Scargle periodograms calculated from $V_{ma}(left)_x$ and $h_{max}(right)$ at five latitudes. The red lines denote the 99% confidence levels.

The periodograms of peak emission and peak height show that in addition to the 183-, 365-, 771- and 3965-day periodic features, there is also a 120-day periodic feature, which we have not analysed in this paper for the 120-day periodic feature.

Lines 181 to 200 – Any new manuscript should include explanations of the origins of the AO, SAO, and QBO, and how these influence the OH emissions. In particular, the AO should be primarily driven by the annual variation of earth-sun distance. So there is an annual cycle of solar radiation along with varying solar radiation on an 11 year cycle. Does any of the 11 year cycle 'alias' into the annual cycle? Could the Fourier techniques mentioned above help sort out different cyclic variations?

Theoretical studies generally agree that atomic oxygen between 80 and 100 km is controlled largely by downward transport from the thermosphere, where its abundances are very large and its photochemical destruction rate is slow(Garcia and Solomon, 1985). Huang and Hickey (2007) noted that vertical transport becomes important when species have long lifetimes and small scale heights. Since O has a relatively long chemical lifetime, O is mainly controlled by dynamics and less affected by chemistry. For species like O3 and OH*, which have smaller lifetimes, it is the combination of chemistry and dynamics that leads to much greater variability in these species.

[Figure]

**Figure 2: Contour plots of the zonal atomic oxygen concentrations in the equatorial (0°) zone predicted by SD-WACCM-X are shown. The green line refers to a contour line of O concentration. The white line is the parameter describing the OH airglow; the white line in Figure (a) is I, Z in Figure (b), $V_{max}$ in Figure (c), and $h_{max}$ in Figure (d).**

Figure 2 shows the atomic oxygen concentration provided by the WACCM model. The green line in the figure represents a contour of the atomic oxygen concentration. We can find that the lower the height corresponding to the contour, the stronger the OH emission and the lower the emission height. The vertical transport of atomic oxygen shows a correlation with OH emission. As was noted above, a considerable amount of O is transported downwards,resulting in an increase in O at lower altitudes. Because of an increased downward flux of O, an increased amount of O is available to recombine with the major gases. Consequently, a large amount of $O_3$ is created through the three-body recombination reaction (reaction 2) due to the increased O, and then some of it destroyed in an exothermic reaction (reaction 1) to produce OH*. Since OH* is mainly produced through reaction 1, it is not surprising to find a large net increase in the OH* number density due to the increased loss of $O_3$ via reaction 1. The increase in OH* is a direct result of the increase in the amount of O3 reacting in reaction 1 and an indirect result of the increase in O transported vertically. By using a spectral full-wave dynamical model and a 2-D OH chemistry model, Huang and Hickey (2007) investigated the latitudinal variations of the wave effects on the minor species in the OH chemistry in the mesosphere/lower thermosphere region. They attributed the increase of the O number density at lower altitudes is due to wave transport, whereas the increase in the number densities of $O_3$ and OH* is due to the combination of wave transport and chemical effects with chemical effects dominating. They suggested that vertical wave transport of O coupling with chemistry could enhance the OH* volume emission rate because of a significant increase in the number density of the OH*. Therefore, vertical motion of O coupling with chemistry does seem to play a significant role in the enhancement of the OH* volume emission rate. They noted that the vertical motion of the species could be due to the action of gravity waves or to other types of waves, such as the tidal motion shown in Zhang and Shepherd (1999).

$$O_3 + \text{H} \xrightarrow{k1} OH(v \leq 9) + O_2 \tag{1}$$

$$O + O_2 + M \xrightarrow{k_{O+O_2+M}} O_3 + M \tag{2}$$

[Figure]

**Figure 3: (a)The wavenumber-period spectrum of SABER Temperature observations at 85 km and equator during day 152-212 of 2015. (b) The spatial structure of the DW1 amplitude on day 180 of 2015. (c) Temporal structure of the DW1 amplitude from 2003 to 2021. (d) Latitude-time distribution of the DW1 amplitude.**

We calculated the amplitude of the 85 km diurnal tide using temperature data from SABER observations, choosing 85 km because the peak height is near 85 km. We analyse the tidal amplitude variations in correlation with the vertical transport of atomic oxygen, and there is SAO in both the tidal amplitude and the vertical transport of atomic oxygen, which may account for the presence of SAO in the OH emission.

Line 240-250 – please explain physically how the QBO in the stratosphere modulates the OH emission in the mesosphere.

QBO is present in both the tidal amplitude and the vertical transport of atomic oxygen, and stratospheric QBO affects the vertical transport of atomic oxygen in the same way that they affect the variation of the tidal amplitude. The increase in OH* is a direct result of the increase in the amount of O3 reacting in reaction 1 and an indirect result of the increase in O transported vertically.

Line 259-270. SABER has temperature, atomic oxygen, atomic hydrogen, and ozone data. These could be analyzed in concert with the OH data and WACCM model results to produce a complete picture of the relative importance of temperature, chemistry, and dynamics in producing the observed variations in the OH emission.

[Figure]

**Figure 4: (a) Time-latitude distributions of ozone mixing ratio near 85 km. (b) Time-latitude distributions of temperature near 85 km. (c) Time-latitude distributions of H mixing ratio near 85 km. (d) Time-latitude distributions of heights corresponding to O concentration contours.**

The ozone, temperature, and atomic hydrogen data from SABER were analyzed in concert with the atomic oxygen data from the WACCM model to provide a comprehensive understanding of the relative importance of temperature, chemistry, and dynamics in generating the observed variations in OH emission. We use a rather crude way of making comparisons by fitting the above quantities with the variation of the corresponding period such as SAO. The R2 statistic obtained from the fit indicates how well the relationship is explained. We believe that the more the variation of the corresponding period explains a quantity (the closer the R2 is to 1), the more that quantity plays a role in the appearance of that periodic variation in OH emission.

$$f = f_0 + a * \cos\left(\frac{2\pi}{183(day)}(t - t_{SAO})\right), \tag{3}$$

$$f = f_0 + a * \cos\left(\frac{2\pi}{365(day)}(t - t_{AO})\right), \tag{4}$$

As shown in Figure 4, (a), (b), and (c) demonstrate the time-latitude distributions of ozone, temperature, and atomic hydrogen near 85 km observed by SABER, respectively. (d) shows the height-time latitude distribution corresponding to the atomic oxygen concentration contour of the WACCM model. We fit the above quantities nonlinearly by least squares using the functions in Equation 3, respectively, and the fitted coefficients of determination (R-square) are shown in Figure 5(a).

We use O3 and H to denote chemical effects and the height of the O concentration contours to denote dynamic effects. As can be seen in Figure 5(a), the $R^2$ of both O3 and O in the equatorial region is very close to 1, which is consistent with our earlier interpretation that the vertical transport of atomic oxygen produces a large amount of O3 through Reaction 8, and then some of the O3 is destroyed in an exothermic reaction (Reaction 1) to produce OH*. The semi-annual oscillations of the OH airglow emission are the

result of the coupling of the vertical transport of atomic oxygen and chemical effects. Equation 4 was used to analyse the AO in the above quantities and the results are shown in Figure 5(b). Similar to OH emission, AO dominates near 50°N/S. In the results, the $R^2$ of the fitted temperature is close to 1, indicating that temperature is likely to be an important factor in the production of OH emission AO. In addition, Marsh et al. (2006) mention that AO in the higher latitude OH emission is consistent with transport by the downward component of the mean meridional circulation, which brings air rich in atomic oxygen from the lower thermosphere into the mesopause region.

[Figure]

**Figure 5: (a) The $R^2$ of the fitting results for each quantity in Figure10 using SAO. (b) The $R^2$ of the fitting results for each quantity in Figure10 using AO.**

To facilitate the analysis of the relationship between OH airglow emission and stratospheric QBO, We choose a 365-day sliding temporal window with a step of 1 day to smooth OH emission rates and heights, which avoids the effect of seasonal variations on the analysis of the results. The smoothed results are shown in Figure 12, indicating the vertical integrated emission rate, centroid altitude, peak emission rate and peak height, respectively.

[Figure]

**Figure 6: (a)Latitude-time distribution of the vertically integrated emission rate I. (b)Latitude-time distribution of the centroid altitude Z. (c)Latitude-time distribution of peak emission rate $V_{max}$. (d)Latitude-time distribution of peak height $h_{max}$. The white line refers to the daily average of the equatorial stratospheric zonal winds, smoothed like the four parameters of the OH airglow. A positive wind speed indicates an eastward wind.**

In order to analyse whether the QBO signal in the OH emission is tidally correlated, we first smoothed the tidal amplitude in Figure 4(d) as in Figure 6, and the result is shown in Figure 7(a). The QBO signal is also shown in the diurnal tidal amplitude. Since QBO modulation of OH emission is also tidally correlated, whether it is observed in the vertical transport of atomic oxygen as in SAO is what we want to know. To explore the relative importance of temperature, chemistry, and dynamics on the QBO signal of OH emission, we performed the same smoothing with a window size of 365 days for the values in Fig. 10. A linear fit was then performed using Equation 5, and the $R^2$ of the fit is shown in Figure 7(b).

$$f = A + B * QBO \ , \tag{5}$$

where QBO represents the zonal wind data at 11 hPa in Singapore. At some latitudes, the R2 value when fitting H is larger than O. Xu et al. (2009) mentioned that SABER measured [O] and [H], but their errors are large, so we only use the SABER observation of [H] as a simple reference and not as the main analysis. In addition, at the equator, where the QBO signal is strongest, the R2 value is largest for O. The weak QBO signals for O3, H, and temperature suggest that dynamical processes play a more important role in producing the QBO signal for OH emission compared to temperature and chemical effects.

[Figure]

**Figure 7: (a) Latitudinal-time distribution of the 85km diurnal tide amplitude. (b) O3, temperature, H and atomic contours of oxygen concentrations corresponding to heights associated with OH airglow emission. After fitting these quantities using the Singapore wind field, the $R^2$ statistics for different latitudes are shown in Fig.**

Assuming that O3, temperature, H and O mentioned in Figure 10 are all linearly related to solar irradiance, we can use linear regression to calculate their response to solar activity and thus analyse which process dominates. A linear approximate regression model is used to describe the relationship between these quantities and solar irradiance:

$$f = A + B * Solar \ , \tag{12}$$

where Solar stands for solar irradiance.

Figure 8 shows the $R^2$ statistics for each latitude after fitting using solar irradiance. The comparison reveals that O3 has the largest R2 statistic, indicating that O3 (ozone-hydrogen reaction) has the largest relative role in the modulation of OH emission by solar activity. Tang et al. (2018) investigated the correlation between ozone concentration and solar activity using broadband emission radiometric atmospheric sounding (SABER) measurements and the 10.7-centimetre solar radiation flux (F10.7) dataset. He noted that O3 is determined by many factors under the radiative, chemical, and dynamical processes. With the increasing of UV irradiance, the solar activity index F10.7 increase, the production of ion pairs by photoionization (N2 and O2) and the production of atoms by O2 photodissociation increase, and thus, the abundance of O atoms vary with solar activity. The photochemical formation of ozone depends on solar activity. These may explain the reason that the interannual variation of O3 and [O] in mesopause are consistent with the 11-year solar cycle. The equatorial regions have $R^2$ statistics that are significantly smaller than those of other latitudinal regions, and those of the southern hemisphere are larger than those of the northern hemisphere. The response of OH airglow emission to solar activity shows the same pattern of latitudinal variation. The chemical reaction coefficient $k = 1.4 \times 10^{-10} \exp(-470/T) b(v)$ (Makhlouf et al., 1995) in reaction (1) where b(v) is positive. Temperature also has an effect on the emission of OH airglow. Marsh et al. (2006) compared SABER observations with a three-dimensional chemical transport model and found that most of the change in emissions was due to changes in ozone. Vertical transport of atomic oxygen is not the most important in the modulation

of OH emission by solar activity. However, the $R^2$ statistic at all latitudes is greater than 0.6, indicating that the vertical transport of atomic oxygen still plays an important role in the modulation of OH emission by solar activity. Therefore, we believe that chemical effects play the most important role in the modulation of OH emission by solar activity.

[Figure]

**Figure 8: O3, temperature, H and atomic contours of oxygen concentrations corresponding to heights associated with OH airglow emission. After fitting these quantities using solar irradiance, the $R^2$ statistics for different latitudes are shown in Fig.**

Line 300- Instead of using the F10.7 proxy, it is suggested to use the actual solar irradiance measured by the SORCE and SEE instruments over the past 20-plus years. Focus on the wavelength regions that drive most of the heating in the mesopause region. This may provide a much better result than using F10.7.

We have used solar irradiance to represent the intensity of solar activity, replacing the previous F10.7. The figure below shows the variation of the mean solar irradiance for wavelengths from 120.5 nm to 150.5 nm compared to the variation of F10.7. We chose a sliding time window of 365 days with a step size of 1 day to smooth the solar irradiance and F10.7.

[Figure]

Figure 8: Solar irradiance vs. F10.7. The black line shows the change in mean solar irradiance from 120 nm to 150.5 nm and the red line shows the change in F10.7.

**References**

Garcia, R. R. and Solomon, S.: THE EFFECT OF BREAKING GRAVITY-WAVES ON THE DYNAMICS AND CHEMICAL-COMPOSITION OF THE MESOSPHERE AND LOWER THERMOSPHERE, J. Geophys. Res.-Atmos., 90, 3850-3868, 10.1029/JD090iD02p03850, 1985.

Huang, T. Y. and Hickey, M.: On the latitudinal variations of the non-periodic response of minor species induced by a dissipative gravity-wave packet in the MLT region, J. Atmos. Sol.-Terr. Phys., 69, 741-757, 10.1016/j.jastp.2007.01.011, 2007.

Makhlouf, U. B., Picard, R. H., and Winick, J. R.: PHOTOCHEMICAL-DYNAMICAL MODELING OF THE MEASURED RESPONSE OF AIRGLOW TO GRAVITY-WAVES .1. BASIC MODEL FOR OH AIRGLOW, J. Geophys. Res.-Atmos., 100, 11289-11311, 10.1029/94jd03327, 1995.

Marsh, D. R., Smith, A. K., Mlynczak, M. G., and Russell, J. M.: SABER observations of the OH Meinel airglow variability near the mesopause, J. Geophys. Res-Space Phys., 111, 14, 10.1029/2005ja011451, 2006.

Tang, C., Wu, B., Wei, Y., Qing, C., Dai, C., Li, J., and Wei, H.: The Responses of Ozone Density to Solar Activity in the Mesopause Region and the Mutual Relationship Based on SABER Measurements During 2002–2016, Journal of Geophysical Research: Space Physics, 123, 3039-3049, https://doi.org/10.1002/2017JA025126, 2018.

Xu, J., Smith, A. K., Liu, H.-L., Yuan, W., Wu, Q., Jiang, G., Mlynczak, M. G., Russell III, J. M., and Franke, S. J.: Seasonal and quasi-biennial variations in the migrating diurnal tide observed by Thermosphere, Ionosphere, Mesosphere, Energetics and Dynamics (TIMED), Journal of Geophysical Research: Atmospheres, 114, https://doi.org/10.1029/2008JD011298, 2009.

Zhang, S. P. P. and Shepherd, G. G.: The influence of the diurnal tide on the O(S-1) and OH emission rates observed by WINDII on UARS, Geophys. Res. Lett., 26, 529-532, 10.1029/1999gl900033, 1999.

---

## Author Comment (AC2)

We thank the reviewers and editors for their constructive comments on our manuscript. The manuscript is revised thoroughly by considering all the comments. Our responses to every comment are listed below with blue.

**Response to Anonymous Referee 2**

**General comments:**

This manuscript deals with the variability in OH emission altitude and OH peak emission rate over a period of more than 20 years based on limb emission measurements with the SABER instrument on TIMED. Different effects and impacts are investigated, e.g. seasonal variations, the effects of the QBO and solar cycle variations. Such a study is in principle of relevance for the mesosphere/mesopause community, because many groups employ ground-based measurements of OH emissions to retrieve mesopause temperature and knowledge about the variability of the OH emission height is important. I have several major and many minor concerns regarding this manuscript, however, and believe that at least a major revision is required before the paper should be accepted in ACP.

I briefly mention my major concerns first, followed by specific comments. It is not really clear, what the new aspects of this study are. It seems to me that all of the shown effects have already been reported in earlier studies. If not, please highlight the new results explicitly. The discussions and arguments often quite weak and it is not clear, what is based on speculation and what on – perhaps – earlier studies, particularly in terms of the role of tides. In addition, the authors chose to analyse variations in peak emission rate and peak altitude. One can certainly do that, but there are other quantities that would be more valuable for the ground-based OH community. I think you should not only analyze peak emission rate, but also the vertically integrated emission rate. The latter is much more important for the many ground-based observers. And this can be done quite easily. You can certainly keep the results on peak emission rate, but add results on the vertically integrated emission rate. Also, the centroid altitude, i.e. altitude weighted by the emission rate profile, would also be a good quantity and is in my opinion more useful than the peak altitude. Finally, there are many linguistical issues, incomplete sentences or sentences, whose meaning is unclear.

**Specific comments:**

Title: "and long-term trend"

Long-term trends are not discussed at all in this paper, as far as I can tell. Please remove this from the title.
We have changed the title to SAO, AO, QBO, and Long-term variation of the OH airglow emission. The word "trend" has also been corrected where it appears in the text.

Line 10: "The results show similar latitudinal variations in the semiannual oscillation (SAO) and

annual oscillation (AO) of peak emission rate and peak height: the amplitude of SAO is greatest in equatorial regions and AO is greatest in mid-latitudes."

I don't think this statement is correct and it probably does not convey the intended meaning, e.g. the latitudinal variation of peak height and peak emission rate are anti-correlated.

The intended meaning is to say that peak emissivity and peak altitude are inversely proportional. At the equator, the SAO amplitude is greatest for both. At higher latitudes, the AO amplitude is greater than the SAO amplitude for both.

Line 14: "The QBO in OH airglow is consistent with the phase variation of the QBO in the tropical lower stratosphere (30km), which is also consistent with the phase variation of the QBO in the migrating diurnal tide."

This is quite a vague statement and I'm not sure it is correct. What specifically is consistent between the variations? Is the QBO-effect on the migrating diurnal tide really understood? Statements of this kind appear throughout the manuscript, but it is unclear, whether this is well established (are there earlier studies? If yes, they should be cited) or speculation? The current manuscript does certainly not provide any evidence that this is the mechanism driving the QBO signature in OH emissions.

The article does lack clarity in describing how tides affect OH emission, and we have added to that section. The section is as follows:
Theoretical studies generally agree that atomic oxygen between 80 and 100 km is controlled largely by downward transport from the thermosphere, where its abundances are very large and its photochemical destruction rate is slow(Garcia and Solomon, 1985). Huang and Hickey (2007) noted that vertical transport becomes important when species have long lifetimes and small scale heights. Since O has a relatively long chemical lifetime, O is mainly controlled by dynamics and less affected by chemistry. For species like O3 and OH*, which have smaller lifetimes, it is the combination of chemistry and dynamics that leads to much greater variability in these species.

[Figure]

Figure 1: Contour plots of the zonal atomic oxygen concentrations in the equatorial (0°) zone predicted by SD-WACCM-X are shown. The green line refers to a contour line of O concentration. The white line is the

**parameter describing the OH airglow; the white line in Figure (a) is I, Z in Figure (b), $V_{max}$ in Figure (c), and $h_{max}$ in Figure (d).**

Figure 1 shows the atomic oxygen concentration provided by the WACCM model. The green line in the figure represents a contour of the atomic oxygen concentration. We can find that the lower the height corresponding to the contour, the stronger the OH emission and the lower the emission height. The vertical transport of atomic oxygen shows a correlation with OH emission. As was noted above, a considerable amount of O is transported downwards,resulting in an increase in O at lower altitudes. Because of an increased downward flux of O, an increased amount of O is available to recombine with the major gases. Consequently, a large amount of $O_3$ is created through the three-body recombination reaction (reaction 2) due to the increased O, and then some of it destroyed in an exothermic reaction (reaction 1) to produce OH*. Since OH* is mainly produced through reaction 1, it is not surprising to find a large net increase in the OH* number density due to the increased loss of $O_3$ via reaction 1. The increase in OH* is a direct result of the increase in the amount of O3 reacting in reaction 1 and an indirect result of the increase in O transported vertically. By using a spectral full-wave dynamical model and a 2-D OH chemistry model, Huang and Hickey (2007) investigated the latitudinal variations of the wave effects on the minor species in the OH chemistry in the mesosphere/lower thermosphere region. They attributed the increase of the O number density at lower altitudes is due to wave transport, whereas the increase in the number densities of $O_3$ and OH* is due to the combination of wave transport and chemical effects with chemical effects dominating. They suggested that vertical wave transport of O coupling with chemistry could enhance the OH* volume emission rate because of a significant increase in the number density of the OH*. Therefore, vertical motion of O coupling with chemistry does seem to play a significant role in the enhancement of the OH* volume emission rate. They noted that the vertical motion of the species could be due to the action of gravity waves or to other types of waves, such as the tidal motion shown in Zhang and Shepherd (1999).

$$O_3 + \text{H} \xrightarrow{k1} OH(v \leq 9) + O_2 \tag{1}$$

$$O + O_2 + M \xrightarrow{k_{O+O_2+M}} O_3 + M \tag{2}$$

[Figure]

**Figure 2: (a)The wavenumber-period spectrum of SABER Temperature observations at 85 km and equator during day 152-212 of 2015. (b) The spatial structure of the DW1 amplitude on day 180 of 2015. (c) Temporal structure of the DW1 amplitude from 2003 to 2021. (d) Latitude-time distribution of the DW1 amplitude.**

We calculated the amplitude of the 85 km diurnal tide using temperature data from SABER observations, choosing 85 km because the peak height is near 85 km. We analyse the tidal amplitude variations in correlation with the vertical transport of atomic oxygen, and there is SAO in both the tidal amplitude and the vertical transport of atomic oxygen, which may account for the presence of SAO in the OH emission. There are latitudes where the tidal amplitude has a large correlation coefficient with the vertical transport of atomic oxygen, and the SAO amplitude of OH emission is also large at these latitudes. So we speculate that tidal variations are correlated with variations in OH emission. QBO is also present in the tidal amplitude, and QBO modulates OH emission, possibly indirectly by affecting the vertical transport of atomic oxygen and thus OH emission.O3 is involved in the reaction to produce OH*, and the production of O3 is correlated with atomic oxygen (reaction 2).

Line 16: "As an important kinetic process"

I would call not tides a kinetic process. They are a "dynamical process"
Kinetic process has been changed to dynamical process.

Line 16: "we suggest that the tides play an important role in the modulation of the OH airglow by the QBO"
Again, the paper provides no evidence that this is the main mechanism.

To facilitate the analysis of the relationship between OH airglow emission and stratospheric QBO, We choose a 365-day sliding temporal window with a step of 1 day to smooth OH emission rates and heights, which avoids the effect of seasonal variations on the analysis of the results. The smoothed results are shown in Figure 3, indicating the vertical integrated emission rate, centroid altitude, peak emission rate and peak height, respectively.

[Figure]

**Figure 3: (a)Latitude-time distribution of the vertically integrated emission rate I. (b)Latitude-time**

distribution of the centroid altitude Z. (c)Latitude-time distribution of peak emission rate $V_{max}$. (d)Latitude-time distribution of peak height $h_{max}$. The white line refers to the daily average of the equatorial stratospheric zonal winds, smoothed like the four parameters of the OH airglow. A positive wind speed indicates an eastward wind.

The ozone, temperature, and atomic hydrogen data from SABER were analyzed in concert with the atomic oxygen data from the WACCM model to provide a comprehensive understanding of the relative importance of temperature, chemistry, and dynamics in generating the observed variations in OH emission. We use a rather crude way of making comparisons by fitting the above quantities with the variation of the corresponding period such as SAO. The R2 statistic obtained from the fit indicates how well the relationship is explained. We believe that the more the variation of the corresponding period explains a quantity (the closer the R2 is to 1), the more that quantity plays a role in the appearance of that periodic variation in OH emission.

In order to analyse whether the QBO signal in the OH emission is tidally correlated, we first smoothed the tidal amplitude in Figure 2(d) as in Figure 6, and the result is shown in Figure 4(a). The QBO signal is also shown in the diurnal tidal amplitude. Since QBO modulation of OH emission is also tidally correlated, whether it is observed in the vertical transport of atomic oxygen as in SAO is what we want to know. To explore the relative importance of temperature, chemistry, and dynamics on the QBO signal of OH emission, we performed the same smoothing with a window size of 365 days for the values in Fig. 10. A linear fit was then performed using Equation 3, and the $R^2$ of the fit is shown in Figure 4(b).

$$f = A + B * QBO \ , \tag{3}$$

where QBO represents the zonal wind data at 11 hPa in Singapore. At some latitudes, the R2 value when fitting H is larger than O. Xu et al. (2009) mentioned that SABER measured [O] and [H], but their errors are large, so we only use the SABER observation of [H] as a simple reference and not as the main analysis. In addition, at the equator, where the QBO signal is strongest, the R2 value is largest for O. The weak QBO signals for O3, H, and temperature suggest that dynamical processes play a more important role in producing the QBO signal for OH emission compared to temperature and chemical effects.

[Figure]

Figure 4 (a) Latitudinal-time distribution of the 85km diurnal tide amplitude. (b) O3, temperature, H and atomic contours of oxygen concentrations corresponding to heights associated with OH airglow emission. After fitting these quantities using the Singapore wind field, the $R^2$ statistics for different latitudes are shown in Fig.

Line 23: "Airglow is the product of photochemical processes in the middle and upper atmosphere, and its radiation intensity is related to atmospheric temperature and atmospheric density."

This is very general and vague. I suggest deleting the second part of the statement or mentioning specific processes.

The energy required for the airglow emission is supplied by solar electromagnetic radiation. Through direct and indirect action, solar radiation excites atoms, molecules and ions in the upper atmosphere to higher energy states, and activates particles to emit photons when they transition from high energy states to lower energy states, i.e., generating airglow. The hydroxyl (OH) vibration-rotational band is generated by the radiation transition between the vibrational energy levels of the ground state of the OH electron. The source of its production mainly comes from the chemical reaction between hydrogen atoms and ozone:

$$O_3 + H \xrightarrow{k1} OH(v \leq 9) + O_2$$                ,

(1)

The temporal and spatial distribution of airglow is modulated by various atmospheric dynamical processes such as atmospheric gravity, tidal and planetary waves. As radiation from the atmosphere itself, the airglow carries a wealth of information about the middle and upper atmosphere and can be used as a medium for the study of various dynamical and photochemical processes in the middle and upper atmosphere. The emission rate and the height of emission of the OH airglow are clearly indicative of atmospheric changes. One can imagine that since the $H + O_3$ reaction is the largest heating source in the mesopause region, changes in the intensity and location of OH emission imply that the energetics of the mesopause is greatly altered. Investigating how the emission rate and intensity of OH emission changes, as well as probing what role temperature, chemistry, and dynamics play and play in this, is an important goal of the research in this paper.

Line 35: "observed strong semiannual" -> "observed a strong semiannual"
Already modified.

Line 38: "with a maximum at the equinoxes and a minimum at the solstices" -> "with maxima at the equinoxes and minima at the solstices"
Already modified.

Line 39: "URAS satellite" -> "UARS satellite"
Already modified.

Line 47: "mesopause thermometer (MTM)" -> "Mesospheric Temperature Mapper (MTM)"
Already modified.

Line 48: "variations of O(1S)" -> "variations of O(1S) green line"   (there is also a UV line originating from the 1S state of O.

Already modified.

Line 54: "With increasing latitude, annual oscillation dominates at higher latitudes."
Please rephrase, sentence logic suboptimal.
At higher latitudes, the amplitude of the AO is greater than that of the SAO.
Line 57: "has made great progress." -> "great progress has been made"
Already modified.

Line 60: "Batista et al. (1994) found a positive correlation between …"
Please mention how long their time series was.
Batista et al. (1994) found a positive correlation between the OH(9,4) airglow intensity obtained in Brazil (23°S, 45°W) from 1977 to 1986 and the F10.7 index by analysing the relationship between this intensity and the intensity of solar activity.

Line 65: "Pertsev and Perminov (2008) analyzed the response of hydroxyl airglow to solar activity …"
Please mention the OH bands they observed.
Pertsev and Perminov (2008) analyzed the response of OH(6-2) airglow to solar activity using observations from the Zvenigorod Observatory (56° N, 37° E) and found that the response of emission intensity to the variation of F10.7 solar radio flux was 30%-40%/100sfu, with a more significant response in winter compared to summer.

Line 72 on SCIAMACHY measurements: the following reference is also relevant for your study and provides a more in-depth analysis of the variability in SCIAMACHY OH(3-1) and OH(6-2) nightglow measurements:

Teiser, G, and C. von Savigny, Variability of OH(3-1) and OH(6-2) emission altitude and volume emission rate from 2003 to 2011, J. Atmos. Sol.-Terr. Phys., 161, 28 - 42, 2017.
Thank you very much, after reading this reference it does help a lot to analyse in depth the variations in OH emission.

Line 73: "Gao et al. (2016) used the airglow data observed by SABER to analyze the response of NO"

I don't think this is an NO airglow emission?
Already modified.

Line 79: "Yee et al. (1997) analyze three-day observations"

What do you mean by "three-day observations"?
Already modified.

Line 94: "In addition to OH airglow intensity, peak emission rate and peak height are important parameters that can be used to describe the airglow emission rate."

I think that the vertically integrated emission rate is more valuable than the peak emission rate. Ground-based instruments observe the first, not the latter.

We have calculated the vertically integrated emission rate and analyzed its variation.

Line 99: "low thermosphere" -> "lower thermosphere"

Already modified.

Line 108: "An overview is given in Section 4."

Not really clear what the overview is about? Perhaps this sentence can be deleted?

A summary is given in Section 4.

Line 114: "but also the NO OH emissions in the excited state"

? What do you mean by "NO OH emissions"? And "emissions in the excited" state does not make much sense.

but also the volume emission rates for NO, OH and O2.

Line 115: "radiation to come from" -> „coming from" or just „from"

Already modified.

Line 118: "observed by bandpass filters with a central wavelength of 1.6 μm"

This was mentioned in the previous sentence already.

Already modified.

Line 119: "SABER can observe the latitude range from 53° in the winter hemisphere to 83° in the summer hemisphere."

Not sure this is correct? There is a two month yaw cycle, right?

Limb profiles are taken from a circular orbit at 625 km inclined at 74◦ to the equator and cover a latitudinal range from 54◦ S to 82◦ N or 82◦ S to 54◦ N, depending on the phase of the yaw cycle(Russell et al., 1999).

Equation (1): Parentheses missing around argument of the sine function.

Already modified.

Line 127: „in the later paper" -> "in the following" ?

Already modified.

Figure 1 and related 1: I think it would be better to determine centroid altitudes, i.e. altitude weighted by the emission rate profile.

We have calculated the centroid altitudes and analyzed their variations, and for the peak emission rates and emission heights that have been calculated we have chosen to keep them.

Line 135: Please use another section title. The reader does not know yet what LRO means. I addition "low resolution OMNI" is not a suitable title.

For solar activity, we have used the solar irradiance representation measured by SEE (The Solar EUV Experiment). The F10.7 provided by OMNI, which was used previously, does not have the acronym found at URL https://omniweb.gsfc.nasa.gov/html/ow_data.html, where the data are provided.

Line 136: What does OMNI stand for?
Already modified.

Line 146: „The vertical resolution of the data is changed from 0.667°“

Something is wrong here. The vertical resolution cannot be in degrees.
The longitudinal resolution of the data is changed from 0.667° in MERRA and the latitudinal resolution remains the same (0.5°).
Line 147: „Their wind field data“

Unclear, what „their“ refers to.
MERRA-2 wind field data contain up to 80 km of zonal wind data with a temporal resolution of 3 hours.
Line 149: „(0.625°N,103.125°S)“
(0.625°N,103.125°E).
Line 152: „were first averaged daily and latitudinally“

? Do you mean zonal means, i.e. averaging over longitude?

Same line: „The averaged results were averaged by latitude range“

What does this mean?

Same sentence: "and the result obtained by averaging was used as the value of the grid centroid.“

It is unclear to me what "grid centroid“ means here.

$I$, $Z$, $V_{max}$ and $h_{max}$ of each emission profile are then averaged over one day and 5° latitude ranges. One yaw cycle of SABER corresponds to 60 days, i.e. due to the full precession of the instrument during one cycle, this period is required to get a full coverage of local times. For this purpose, we make a sliding window with a time length of 60 days and a sliding step of 1 day, and the combination of data distributed in each sliding window can cover 24 hours of local time. We consider the average value of the data in each sliding window as the data of the day in the center of the window, so that the data of each day meets the 24-hour local time coverage.

Line 154: "cover the global 24h place time“
?? Meaning unclear.
24-hour local time coverage

Line 167: "The maximum value is at the equinox and the minimum value is at the solstice, and the extreme value at the September equinox is larger than that at the March equinox,"

I disagree, your plot shows the opposite behaviour.

The maximum value is at the equinox and the minimum value is at the solstice, and the extreme value at the March equinox is larger than that at the September equinox,

Line 173: "The same phenomenon has been identified in previous studies and is thought to be the role of tides in this"

Sentence incomplete or wrong.

Previous studies have found the same phenomenon and concluded that tides have an effect on SAO in OH emission.

Line 187: "The phase of SAO is delayed from near day 90"

Unclear, what this means or what the intended meaning is.

The point I'm trying to make is that the equatorial region has the first maximum in SAO amplitude, and other latitudes have a later maximum in SAO amplitude than the equatorial one. The expression has been modified: Looking at the phase of the SAO we can see that the maximum value of the amplitude in the equatorial region occurs on the 88th day of the year.

Line 188: "The peak emission rate has its largest amplitude during the equinox, which is consistent with diurnal tides (Burrage et al., 1995)."

I don't think it is that easy. We don't know what local solar time the measurements are made at the equinoxes! Also, the statement is not really precise. In what way do you see consistency here. What specific mechanism do you have in mind?

There are explanations of how the tides affect OH emission in earlier replies and in the paper.

Line 189: "The peak emission rate is associated with diurnal tides and ist seasonality is likely to be caused by the seasonal variation of diurnal tides."

Are there any references to back this up? Also, "the peak emission rate is associated with diurnal tides" doesn't really make sense.

There are explanations of how the tides affect OH emission in earlier replies and in the paper.

Line 196: "is greatest on day 183 of the first year"

?? I don't really understand this. The AO has a period of 1 year, i.e. one maximum every year on the same day of the year??

We previously calculated a phase range of -π to π, which is the second half of the first year and the first half of the second year, and we have now converted the phase range to [0,366].

Line 197: "The amplitude of the 50°S reaches its maximum on day 140 of the second year and the 50°N reaches its maximum on day 337 of the first year"

See last point.

We previously calculated a phase range of -π to π, which is the second half of the first year and the first half of the second year, and we have now converted the phase range to [0,366].

Line 199: "and annual oscillation is more frequent at higher latitudes"

"more frequent" is not a good choice of words in this context

The AO amplitude is very small at low latitudes, and the AO is dominant at higher latitudes.

Line 200: "We note the semiannual and annual variations in OH airglow intensity provided by Reid et al. (2014), who analyzed filter photometer measurements at Buckland Park"

If you would determine vertically integrated emission rates, you could do an apples-to-apples comparison with Reid et al.

We have calculated and compared.

Line 204: "and the trend of the phase and amplitude change of the peak height"

What does trend mean here? Please specify or replace? Trend is usually/often the "linear trend"

What we are trying to convey is that as the latitude changes, the amplitude and phase appear to change accordingly, i.e. latitude dependence.

Line 211: "reaches its maximum amplitude on day 2 of the second year and the 50°N reaches its maximum on day 178 of the second year."

Again, the AO has a period of 1 year; why do you have to specify year 1/year 2?

We previously calculated a phase range of -π to π, which is the second half of the first year and the first half of the second year, and we have now converted the phase range to [0,366].

Line 222: "The atomic oxygen distribution derived by Russell and Lowe (2003) using the hydroxyl emission and oxygen green line emission observed by WINDII exploits precisely the relationship between OH emission and atomic oxygen."

This is only partly correct. The [O] retrievals from the OI green line do not exploit the relationship between the OH emission and O.

Already modified.

Line 227: "One of the mesospheric quasi-biennial oscillation"

?? Meaning unclear.

The Mesopause Quasi-Biennial Oscillation (MQBO) shows the same altitudinal and latitudinal structure as the Mesopause Semi-Annual Oscillation (MASO) and exhibits a phase relationship with the SQBO

Line 227: "One of the mesospheric quasi-biennial oscillation (MQBO) shows a similar spatial distribution structure to the mesospheric semiannual oscillation"

What spatial distribution (dimensions?) do you mean here?

The Mesopause Quasi-Biennial Oscillation (MQBO) shows the same altitudinal and latitudinal structure as the Mesopause Semi-Annual Oscillation (MASO) and exhibits a phase relationship with the SQBO

Line 229: "In addition, the Christmas Island MF radar (2°N, 130°W) also detects an MQBO with the same phase and the same peak height,"

The same phase and peak height compared to what?

In addition, an MQBO was also detected by the Christmas Island MF radar (2°N, 130°W), with the same phase and the same peak height compared to the MQBO shown by HRDI. Although the amplitude is only half of that shown by HRDI, this observation also confirms the existence of this MQBO.

Line 258: "Xu et al. (2009) analyzed the quasi biennial oscillation of the migrating diurnal tide based on data from TIMED observations."

Which observations, i.e. which atmospheric parameter(s)? And from which instrument? Altitude range?

Xu et al. (2009) analyzed the quasi-biennial oscillation of the migrating diurnal tide based on temperature and wind data from TIMED observations.

SABER and TIDI began making observations of the global temperature, pressure and wind profiles in late January 2002. SABER retrievals give profiles of the global temperature and pressure in the stratosphere, mesosphere and lower thermosphere. TIDI retrievals give neutral horizontal winds in the upper mesosphere and lower thermosphere. In this paper, we use temperature and pressure profiles from SABER version 1.07 from February 2002 to December 2007 and wind data from TIDI (NCAR produced version 0307) from February 2002 to June 2007.

Line 259: "After comparison, we find that the distribution of OH airglow emission at low latitudes is similar to that of the migrating diurnal tide in temperature."

Unclear, how you came to this conclusion? This is a complicated matter and you should describe in detail what you mean.

We calculated the tides using temperatures measured by SABER. As shown in Fig. 2(d), there is a correlation between the change in tidal amplitude near 85 km and the change in OH emission, respectively, in the equatorial region.

[Figure]

Line 258 – 270: The reasoning in this paragraph is not really stringent, the arguments not precise and the underlying mechanisms are not addressed. For example in line 269: "The stratospheric QBO phase variation is consistent with the tides in the MLT region and the airglow emission phase variation." In which way is there a consistency between the two phenomena? This is completely unclear.

There are explanations of how the tides affect OH emission in earlier replies and in the paper.

Line 275: "Figure 6 shows the correlation with latitude,"

No, this figure does not show a correlation with latitude.

The figure shows the variation of the correlation coefficient with latitude.

Line 276: "and stratospheric QBO"

One cannot correlate data with the "stratospheric QBO". You mean the zonal wind at 10/30 hPa above Singapore, right?

Yes, we use the zonal winds at 11hPa in Singapore to represent the stratospheric QBO.

Line 296: "then OH airglow emission is likely to vary with solar activity"

Replace "then" with "the" ... or just "OH airglow emissions are likely to vary.."

the OH airglow emission is likely to vary with solar activity

Line 308: "with an advance in the response of OH airglow emission in 2008,"
What does "advance" mean here specifically? I suggest rephrasing this sentence.

[Figure]

**Figure 5: Annual mean variation of OH airglow emission (black line) and annual mean variation of solar irradiance (red line). (a), (b), (c) and (d) correspond to I, Z, $V_{max}$ and $h_{max}$, respectively.**

As shown in figure 5, solar irradiance was still decreasing in 2008-2009 and OH emissivity showed an enhancement.

Line 314: "Von Savigny (2015) found no clear long-term trends or 11-year solar cycle features in the OH emission height time series."
The above mentioned paper by Teiser & von Savigny (2017) would also be relevant here.
Von Savigny (2015) found no clear long-term trends or 11-year solar cycle features in the OH emission height time series, partly contradicting earlier studies.

Line 320: "although showing an opposite trend to solar activity"

I suggest not using "trend" in this context.
Already modified.

Line 330: "Analysis of the solar cycle dependence on the peak emission rate and its solar response."

Sentence incomplete.
Already deleted.

Same line: "A global annual mean series scatter plot"

Wording not precise.
Already deleted.

Line 334: "To investigate the solar activity dependence of the OH airglow."

Sentence incomplete.
Already deleted.

Lines 330 – 335: these sentences are partly redundant and not in very good shape overall.

Already deleted.

Line 337: "A is a constant and B is the coefficient"

B is also a constant.

Already modified.

Line 362: "-0.13 km/sfu" -> "-0.13 km/(100 sfu)"

Already deleted.

Line 378: "For QBO, we find that the phase variation of OH airglow emission in the equatorial region remains consistent with the stratospheric QBO phase variation."

What exactly does consistency mean here? Is this just: "there is a general connection" or "we understand the underlying physical mechanism"?

An explanation of how tides affect OH emission and how QBO modulates OH emission is provided in previous responses and papers.

Line 380: "mixing rate" -> "mixing ratio" (and concentration is more relevant here than the mixing ratio)

Already modified.

Line 380: "As the stratospheric zonal wind field moves eastward,"

Wording not precise (wind field moves eastward): Also, what altitudes/latitudes does this refer to?

The emission rate is relatively large during westerly phases, e.g. in 2006. Relatively small emission rates during easterly phases, e.g. 2007.

Line 382: "We, therefore, speculate that tides play an important role in the modulation of OH airglow emission by QBO."

Earlier in the paper this was not phrased that carefully, but presented rather as a fact. The arguments in this paragraph are not very convincing.

An explanation of how tides affect OH emission and how QBO modulates OH emission is provided in previous responses and papers.

Figure 9: This figure should be removed in my opinion. The underlying mechanisms are not well explained. If there are earlier papers, please cite them. And: $O^3$ (!) + H -> $OH^*$ + $O^2$ (!!)   is incorrect.

This figure illustrates the mechanism by which SAO, QBO and solar activity regulate OH airglow. Marsh et al. (2006) compared model and observational data and found that most of the variability

in OH airglow emission is caused by changes in the production rate of ozone. At the height of the peak emission, the variation is mainly caused by changes in atomic oxygen due to vertical transport. Atomic oxygen has a relatively long chemical lifetime, and the vertical transport of O coupled to the chemistry affects OH emission. Atomic oxygen produces large amounts of O3 through reactions, and then some of the O3 is destroyed in reactions with H to produce OH*. The vertical transport of atomic oxygen reacts to the seasonality and QBO of the tidal amplitude, and the change of atomic oxygen will cause the change of O3, which in turn affects the OH emission. OH emission is positively correlated with solar activity and emission height is inversely correlated with solar activity. Among the factors affecting OH emission, O3 and temperature are most affected by solar activity, suggesting that chemical processes play a major role in the modulation of OH emission by solar activity. Vertical transport of atomic oxygen also responds to solar activity, but is obviously more affected by tides.

[Figure]

Line 395: "As a result, the latitudinal variations of peak emission rate and height were similar"

?? Why similar? They were anticorrelated, right?
Yes, they are anti-correlated. As the emissivity SAO amplitude gets larger as latitude changes, the emissivity altitude SAO amplitude also gets larger.

Line 401: "appears on day 183 of the first year at 25°N, delayed towards the poles. The maximum amplitude occurs at 50°S on day 140 of the second year and at 50°N on day 337 of the first year"

Same point as above. Why do you need to specify the year here?
Already modified.

Line 404: "The phase of the SAO at peak height"

??
Already modified.

Line 405: "the AO is delayed from 20°S on day 186 of the first year to day 2 of the second year at 50°S and day 178 of the second year at 50°N."

See point above.
Already modified.

Line 411: "so that the phase variation remains consistent with the stratospheric zonal winds"

Again, this is not precise; consistent in which respect?

Already deleted.

Same line: "When the wind field is to the east,"

??

Already modified.

Lines 414 – 419: Are there earlier studies backing this up? Last sentence: How do you know this? No evidence for this has been presented.

An explanation of how tides affect OH emission and how QBO modulates OH emission is provided in previous responses and papers.

Line 422: "while peak OH airglow emission heights show no significant solar cycle variations."

?? Why not, the correlations coefficients (i.e. their absolute values) are quite large.

Yes, they do show relevance and we have made changes.

Next sentence: Please delete it – this is only speculation.

Already deleted.

**References**

Garcia, R. R. and Solomon, S.: THE EFFECT OF BREAKING GRAVITY-WAVES ON THE DYNAMICS AND CHEMICAL-COMPOSITION OF THE MESOSPHERE AND LOWER THERMOSPHERE, J. Geophys. Res.-Atmos., 90, 3850-3868, 10.1029/JD090iD02p03850, 1985.

Huang, T. Y. and Hickey, M.: On the latitudinal variations of the non-periodic response of minor species induced by a dissipative gravity-wave packet in the MLT region, J. Atmos. Sol.-Terr. Phys., 69, 741-757, 10.1016/j.jastp.2007.01.011, 2007.

Russell, J., Mlynczak, M., Gordley, L., Tansock, J., and Esplin, R.: Overview of the SABER experiment and preliminary calibration results, SPIE's International Symposium on Optical Science, Engineering, and Instrumentation, SPIE1999.

Xu, J., Smith, A. K., Liu, H.-L., Yuan, W., Wu, Q., Jiang, G., Mlynczak, M. G., Russell III, J. M., and Franke, S. J.: Seasonal and quasi-biennial variations in the migrating diurnal tide observed by Thermosphere, Ionosphere, Mesosphere, Energetics and Dynamics (TIMED), Journal of Geophysical Research: Atmospheres, 114, https://doi.org/10.1029/2008JD011298, 2009.

Zhang, S. P. P. and Shepherd, G. G.: The influence of the diurnal tide on the O(S-1) and OH emission rates observed by WINDII on UARS, Geophys. Res. Lett., 26, 529-532, 10.1029/1999gl900033, 1999.